# Motoneurons regulate the central pattern generator during drug-induced locomotor-like activity in the neonatal mouse

Melanie Falgairolle[1]*, Joshua G Puhl[2], Avinash Pujala[3], Wenfang Liu[1], Michael J O'Donovan[1]

[1]Developmental Neurobiology Section, National Institute of Neurological Disorders and Stroke, National Institutes of Health, Bethesda, United States; [2]Department of Entomology, University of Minnesota, Saint Paul, United States; [3]Janelia Research Campus, Howard Hughes Medical Institute, Ashburn, United States

**Abstract** Motoneurons are traditionally viewed as the output of the spinal cord that do not influence locomotor rhythmogenesis. We assessed the role of motoneuron firing during ongoing locomotor-like activity in neonatal mice expressing archaerhopsin-3 (Arch), halorhodopsin (eNpHR), or channelrhodopsin-2 (ChR2) in Choline acetyltransferase neurons (ChAT[+]) or Arch in LIM-homeodomain transcription factor $Isl1^+$ neurons. Illumination of the lumbar cord in mice expressing eNpHR or Arch in ChAT[+] or $Isl1^+$ neurons, depressed motoneuron discharge, transiently decreased the frequency, and perturbed the phasing of the locomotor-like rhythm. When the light was turned off motoneuron firing and locomotor frequency both transiently increased. These effects were not due to cholinergic neurotransmission, persisted during partial blockade of gap junctions and were mediated, in part, by AMPAergic transmission. In spinal cords expressing ChR2, illumination increased motoneuron discharge and transiently accelerated the rhythm. We conclude that motoneurons provide feedback to the central pattern generator (CPG) during drug-induced locomotor-like activity.

*For correspondence: melanie.falgairolle@nih.gov

**Competing interests:** The authors declare that no competing interests exist.

## Introduction

Motoneurons, whose primary function is to produce muscle contraction, are among the most widely studied cells of the nervous system (for review see *Stuart et al., 2011*). In addition to their connections with muscle, motoneurons also project within the spinal cord to synapse with inhibitory Renshaw cells (*Renshaw, 1941, 1946*). In turn, Renshaw cells project back to motoneurons forming a negative feedback circuit (*Eccles et al., 1954*). The central effects of motoneuron discharge have been mostly considered in terms of this negative feedback circuit and have included decorrelation of motoneuron activity (*Maltenfort et al., 1998*) and motoneuron gain control (*Hultborn et al., 1979*). However, Renshaw cells also project to each other, to Ia inhibitory interneurons and to ventral spino-cerebellar neurons (*Lundberg and Weight, 1971*; *Jankowska and Hammar, 2013*). Accordingly, motoneuron discharge could have much wider effects on spinal circuit function than has been previously acknowledged. Invertebrate motoneurons are critically involved in rhythmogenesis in several systems including leech swimming (*Hashemzadeh-Gargari and Friesen, 1989*), the crab stomatogastric system (*Weimann et al., 1991*), drosophila larval locomotion (*Matsunaga et al., 2017*) and nematode locomotion (*Chalfie et al., 1985*). Among vertebrates, motoneuron activity can regulate swimming in the adult zebrafish (*Song et al., 2016*), initiates spontaneous rhythmic motor activity in

the developing chick (*Wenner and O'Donovan, 2001*) and mouse (*Hanson and Landmesser, 2003*) spinal cords and regulates fictive vocalization in *Xenopus Laevis* (*Lawton et al., 2017*). In mammals, motoneurons are not thought to be part of the spinal locomotor generator (*Rybak et al., 2015*) although they can modify their output through activation or inhibition of their intrinsic membrane properties (*Hounsgaard et al., 1984*, *1988*). However, in the *in-vitro* neonatal mouse spinal cord, stimulation of motor axons can trigger a bout of locomotor-like activity (*Mentis et al., 2005*). In addition, a brief burst of ventral root stimuli can entrain disinhibited bursting in the neonatal rat (*Machacek and Hochman, 2006*) and mouse spinal cords (*Bonnot et al., 2009*). Furthermore, in the neonatal rat spinal cord, ventral root stimulation can increase the frequency of drug-induced loco-motor activity if noradrenaline is present in the bath (*Machacek and Hochman, 2006*). Collectively, these findings suggest that motoneuron activity can access the circuitry of the locomotor central pattern generator. For these reasons, we decided to investigate whether motoneuron firing could influence the locomotor CPG during locomotor-like activity induced by bath application of NMDA and serotonin (5-HT) in the neonatal mouse spinal cord. To accomplish this, we expressed archaerhodopsin or halorhodopsin - light-gated proton (*Chow et al., 2010*) and chloride (*Zhang et al., 2007*) pumps respectively - that can hyperpolarize neurons or channelrhodopsin, a light-gated channel that can depolarize neurons (*Boyden et al., 2005*) in ChAT$^+$ neurons, and archaerhodopsin in *Isl1*$^+$ neurons. While they are not specific markers of motoneurons, *Isl1* and ChAT are both expressed in motoneurons as well as in other neurons (*Pfaff et al., 1996*; *Bui et al., 2013*). We then used light to modify the firing of motoneurons and established the resultant effects on the drug-induced locomotor rhythm. Some of this work has appeared in abstract form (*Falgairolle et al., 2016*).

## Results

### Experiments in animals expressing archaerhodopsin-3 in cholinergic neurons

We first established which neurons expressed Arch-GFP using immunocytochemistry for GFP and ChAT. As shown in *Figure 1*, all neurons expressing ChAT immunoreactivity expressed the fusion protein Arch-GFP. As expected, GFP expression was seen in motoneurons (*Figure 1C1*), preganglionic autonomic neurons, ChAT$^+$ neurons surrounding the central canal (*Figure 1C2*), dorsal cholinergic neurons (*Figure 1C3*) and, cholinergic interneurons scattered throughout the ventral gray matter in both L1/2 and L5/6 segments (n = 3 for each) and along the length of the lumbar cord (data not shown).

### Effects of green light on the pattern of locomotor-like activity recorded from the ventral roots and individual motoneurons in ChAT-Arch animals

In previous work, it was shown that dopamine blocked the ability of ventral root stimulation to activate the locomotor rhythm (*Humphreys and Whelan, 2012*). For this reason, our experiments were performed using only NMDA and serotonin to produce locomotor-like activity. To activate archaerhodopsin, we used a green light that illuminated most, if not all, of the lumbar spinal cord (see Materials and methods). *Figure 2* shows the effects of the light on the locomotor rhythm in a ChAT-Arch spinal cord. To assess whether the light itself had any effect on the rhythm we repeated the experiments on wild type (WT) cords, which are devoid of opsins. To statistically compare the effects of light on WT cords and on those expressing opsins, we generated time series of the changes in phase, frequency and integrated motoneuron firing for both the experimental and wild type animals. We then performed a bootstrap t-test on the time series (see Materials and methods) and plotted the results with colors coding for the statistically significant differences between the genetically modified and the wild type cords (*Figure 2B–D*). For the ChAT-Arch cords, we found that the frequency of the rhythm was transiently slowed (*Figure 2C*, p<0.0001 at the minimum) and this was accompanied by a transient change in the phase relationships of the activity (*Figure 2B*, p<0.01). Notice that the changes in frequency and phase at the beginning of the light occur just before the onset of the light. This is because the wavelet analysis produces a temporal smearing that depends upon the frequency components in the signal; the lower the frequency the greater the smearing (*Torrence and Compo, 1998*). While the light slightly increased motoneuron firing in the WT cords (*Figure 2—*

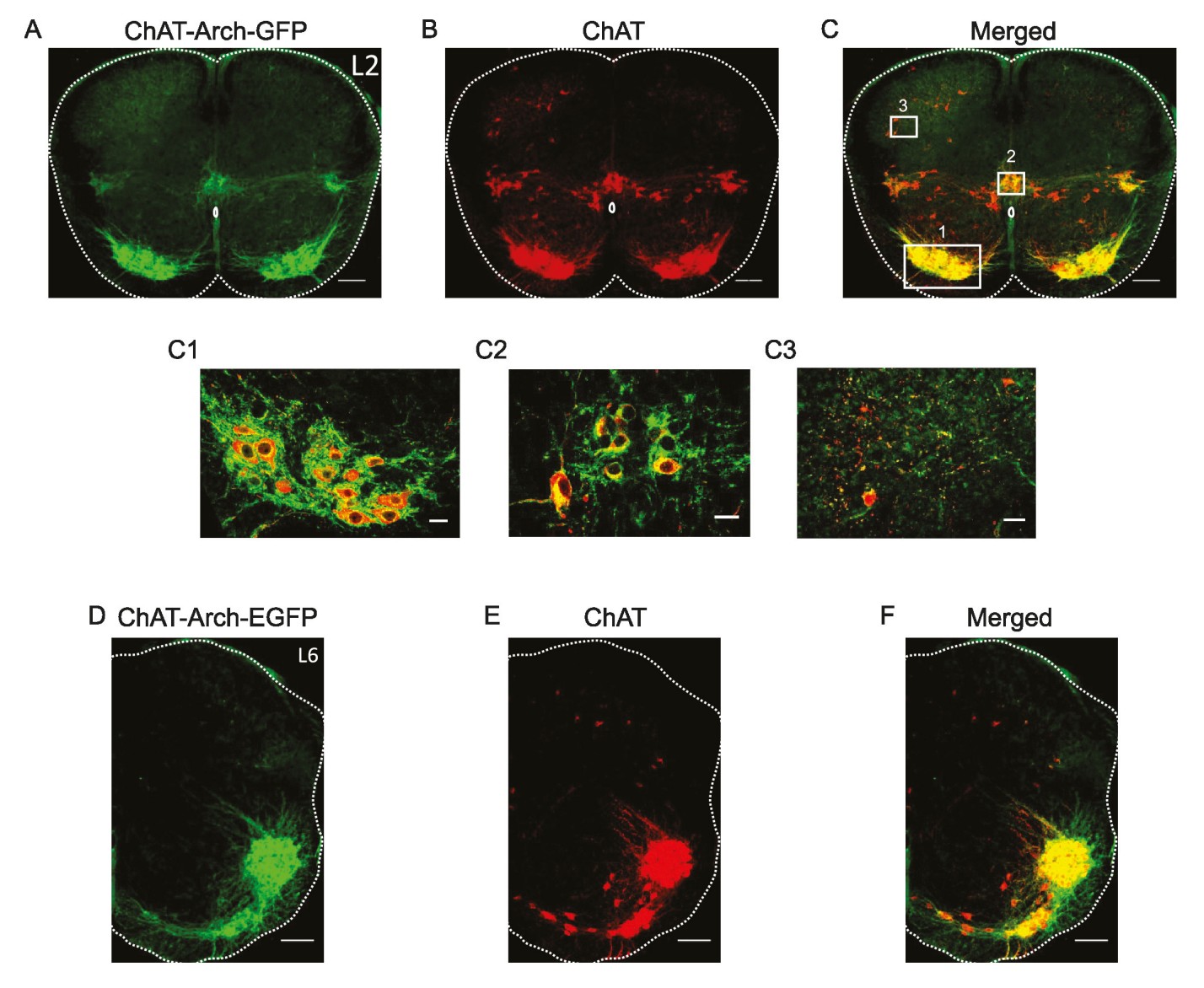

**Figure 1.** Archaerhodopsin is expressed in all ChAT-positive neurons. (A–B–C) Z-stack projection of 10X images (2.05 μm) of a 60 μm section of the L2 segment of a P3 ChAT-Arch mouse spinal cord showing archaerhodopsin (green, **A** and **C**) and ChAT-positive (red, **B** and **C**) neurons and the merged image (**C**). The white scale bars represent 100 μm. Insets from the white rectangles in the merged image show that motoneurons (**C1**), sympathetic pre-ganglionic neurons (**C2**) and dorsal cholinergic neurons (**C3**) all express archaerhodopsin and ChAT (1 μm optical section). The white scale bars measure 20 μm. (D–E–F) 10X z-stack projection (4 μm) of a 60 μm section in the L6 segment of a P3 ChAT-Arch mouse hemi-cord showing archaerhodopsin (green, **D–F**) and ChAT-positive (red, **E–F**) neurons and the merged image (**F**). The white scale bar is 100 μm.

figure supplement 1), it produced a strong inhibition of motoneuron firing in the ChAT-Arch cords (*Figure 2D*, p<0.0001). The inhibition was greater in the extensor-dominated versus the flexor-dominated ventral roots during the 60 s exposure to the light (first 10 s of the light −78.16 ± 7.13%, −63.06 ± 11.19%, Last 10 s of the light: −51.41 ± 6.65%, −40.55 ± 7.94% for extensor and flexor roots respectively, n = 25, 2-way ANOVA, light status: F(3,192) p<0.0001, root F(1,192): p<0.0001, interaction F(3,192): p<0.0001). The frequency decrease in the bilateral flexor roots versus the ipsilateral flexor/extensor roots was not significantly different (p=0.86, 2-way ANOVA, light status: F (3,192) p=0.9934, pair of roots F(1,192): p<0.0001, interaction F(3,192): p=0.8469). The light-induced change in phasing was greatest for the ipsilateral flexor/extensor roots (46.87 ± 33.96°,

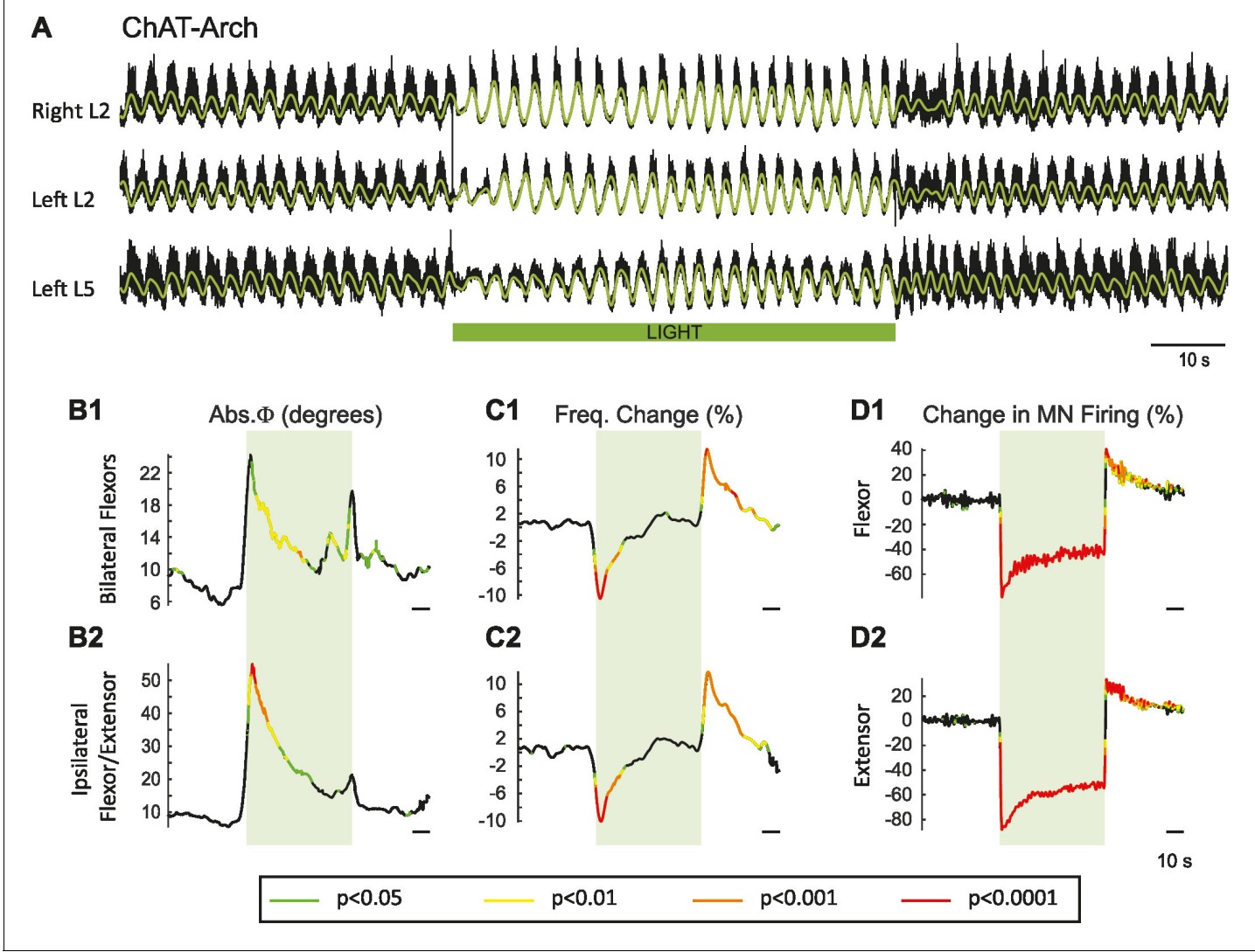

**Figure 2.** Light-induced hyperpolarization of cholinergic neurons transiently decreases the frequency and alters the phasing of drug-induced locomotor-like activity. (**A**) Locomotor-like activity recorded from the right L2 and left L2 and L5 ventral roots (black traces) of a P1 isolated spinal cord of a ChAT-Arch mouse. The superimposed green traces are the slow potentials obtained by low pass filtering the raw signals. Locomotor-like activity was evoked by applying 5 µM NMDA and 10 µM 5-HT. The duration of the light (60 s) is indicated by the green bar. (**B–C**) Time series showing the change in absolute phase (**B**) and frequency (**C**) averaged for all experiments for the bilateral flexor (**B1–C1**) and the ipsilateral flexor/extensor roots (**B2–C2**). (**D**) Change (%) in the averaged integrated ventral root discharge (Change in MN firing) for the ipsilateral flexor (**D1**) and extensor (**D2**) ventral roots. The statistics were obtained using a bootstrap t-test between ChAT-Archaerhodopsin (n = 25) and wild type cords (n = 28) and are color-coded as indicated in the box below the records.

The following figure supplements are available for figure 2:

**Figure supplement 1.** Effect of green light on drug-induced locomotor-like activity in wild type cords.

**Figure supplement 2.** Effect of varying the green light intensity on the integrated firing of the right L2 ventral root and on the frequency of the locomotor-like rhythm in ChAT-Arch cords.

20.07 ± 12.47° ipsilateral and bilateral roots respectively; p<0.0001; n = 25, 2-way ANOVA, light status: F(3,192) p<0.0001, pair of roots F(1,192): p<0.0001, interaction F(3,192): p=0.0002). The frequency of the rhythm returned to control values during the light and then transiently increased when the light was turned off (*Figure 2C*, Last 10 s of the light: 0.7 ± 6.3%, first 10 s after the light: 8.37 ±

5.46%; p<0.0001). Motoneuron firing also increased transiently after the light was turned off (*Figure 2D*, Last 10 s of the light: −40.55 ± 7.95%, first 10 s after the light: 27.16 ± 16.35%; p<0.0001). Similar, but smaller and briefer, light-induced changes were observed in frequency and phase of the rhythm in the WT cords (*Figure 2—figure supplement 1*). These results suggested that the light itself had some non-specific effects on the rhythm and that a component of the changes observed in the ChAT-Arch cords are due to the light probably because of an increase in temperature. We measured the temperature under the light probe, and found that it increased locally by about 2 to 3°C. We then performed experiments to test whether decreasing the intensity of the light would circumvent the problem. We found that, in the ChAT-Arch cords, as the light intensity decreased, the suppression of motoneuron firing was reduced and the changes in the frequency of the locomotor rhythm were smaller and more variable (see *Figure 2—figure supplement 2*). When the light intensity was reduced to 20–25%, motoneuron firing was reduced by ~25–30% (p<0.05) and this was similar whether the cord was illuminated ventrally or dorsally (probably because of light reaching motoneuron dendrites). Therefore, we decided to use the maximum illumination intensity.

## The light-induced changes in frequency and phase are not due to alterations in extracellular or intracellular pH

Recent studies have shown that both local extracellular and intracellular pH are affected by archaerhodopsin activation (*Zeng et al., 2015*; *El-Gaby et al., 2016*; *Mahn et al., 2016*). Because these changes can potentially affect the rhythm (*Beg et al., 2008*; *Zeng et al., 2015*; *Jalalvand et al., 2016*), we repeated the experiments in spinal cords expressing the light-activated chloride pump halorhodopsin (eNpHR; n = 12, *Figure 3A*) and activated the opsin with the same green light. As described for the ChAT-Arch cords, light suppressed the ventral root firing especially in the extensor roots (*Figure 3E*), transiently reduced the frequency, and altered the phasing of the rhythm (*Figure 3C and D*). The light-induced changes in phase were greater in the ipsilateral flexor/extensor roots than in the bilateral flexor roots (*Figure 3C*). After the light was turned off, a rebound in frequency and motoneuron spiking was also observed (*Figure 3D*).

To compare statistically the light-induced changes in the variables between the different groups (WT, ChAT-Arch, ChAT-eNpHR) we calculated their mean value 10 s before light (control), the first 10 s of the light (Start Light), the last 10 s of the light, and the first 10 s after the light (After Light). For the quantitative analysis, we focused on the behavior of the bilateral flexors because the phasing of the flexor/extensor roots was susceptible to light-induced changes in phase caused by the membrane potential becoming lower than the chloride equilibrium potential (see below and *Figure 3—figure supplement 1*).

The frequency of the control rhythm (before the light, *Figure 3B*) was significantly faster in the ChAT-Arch cords compared to the wild type cords (WT: 0.43 ± 0.066 Hz, ChAT-Arch: 0.48 ± 0.077 Hz, p=0.0159) but not in the ChAT-eNpHR cords compared to the wild type cords (0.46 ± 0.06 Hz). During the light, both ChAT-Arch and ChAT-eNpHR cords exhibited significant light-induced decreases in motoneuron firing compared to control levels (p<0.0001) with ChAT-Arch cords being silenced more than ChAT-eNpHR cords (*Figure 3H*; ChAT-Arch: −63.06 ± 11.19%, ChAT-eNpHR: −53.75 ± 12.01%, p=0.0030). A small light-induced increase in motoneuron firing was observed in the wild type cords when the light was turned on (5.62 ± 12.21%, *Figure 2—figure supplement 1D*). We next compared the light-induced changes in frequency (%) to the control frequency (Start Light, *Figure 3G*). Both ChAT-Arch and ChAT-eNpHR cords exhibited significant, transient decreases in frequency (−7.78 ± 3.96% p=<0.0001, −4.46 ± 6.75% p=0.0263 respectively) during the light compared to the control period, whereas the wild type cords did not (−0.69 ± 5.1% p=0.8959). During the first 10 s of the light, the mean ChAT-Arch and ChAT-eNpHR frequencies were statistically different from the wild type (p<0.0001, p=0.0039; respectively) but they were not statistically different from each other (p=0.0936).

We then compared the changes (%) in motoneuron firing and the frequency of the rhythm 10 s immediately following the cessation of the light to the values during the last 10 s of the light (After Light, *Figure 3G*). Both ChAT-Arch and ChAT-eNpHR cords showed a significant increase in motoneuron firing (27.16 ± 16.35%, p<0.0001, 13.16 ± 5.56%, p<0.0001 respectively, data not shown), whereas only the ChAT-Arch cords showed a significant increase in frequency when compared to the frequency during the last 10 s of the light (*Figure 3G*; ChAT-Arch: 8.37 ± 8.46 % p<0.0001, ChAT-eNpHR: 7.21 ± 5.01%, p=0.4331). However, the post-light increase was significantly different

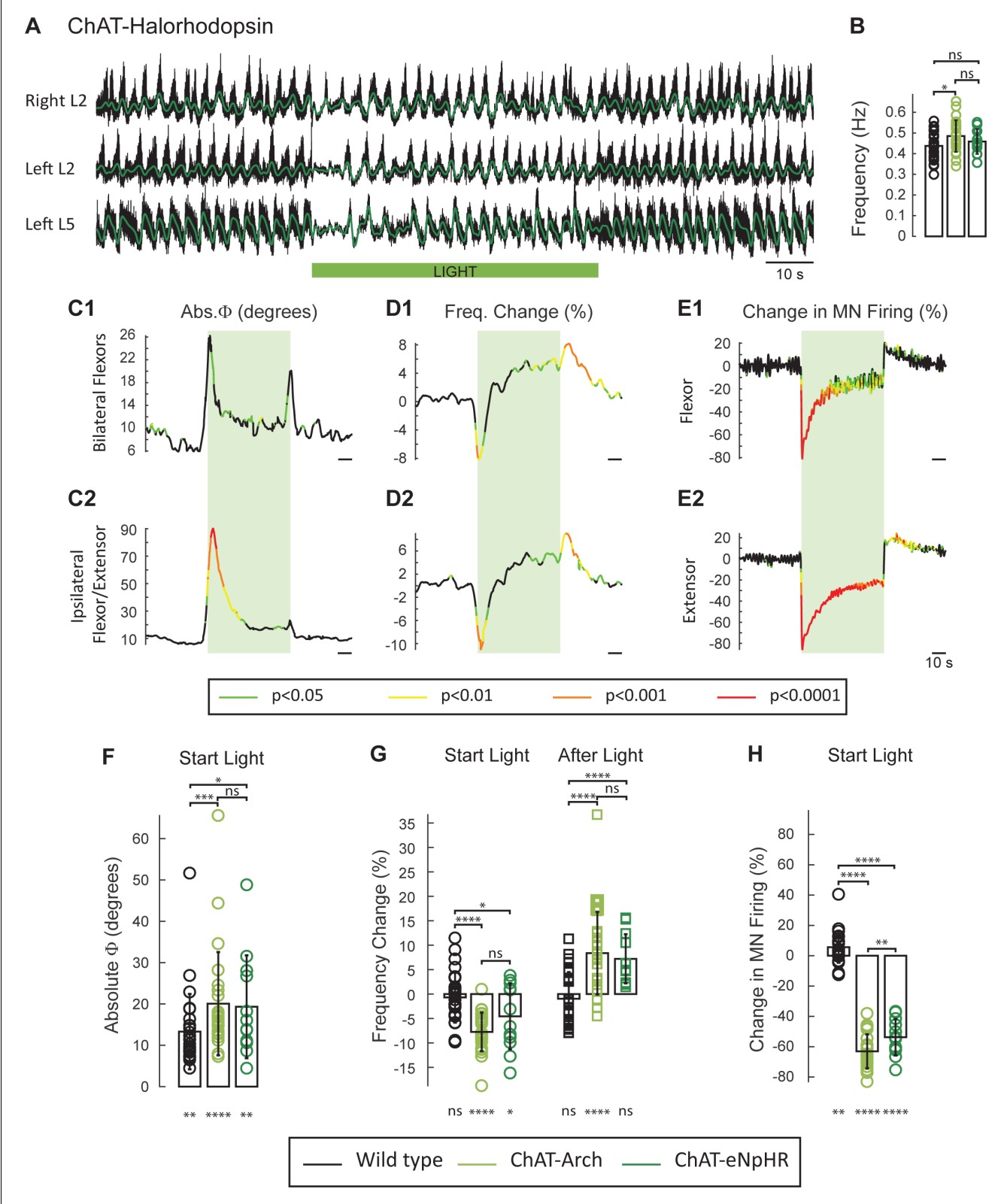

**Figure 3.** The light-induced decrease in the frequency of the rhythm and the phase changes are not due to changes in pH. (**A**) Locomotor-like activity recorded from the right and left L2 and the left L5 ventral roots (black traces) in a P1 ChAT-eNpHR animal. The superimposed green traces are the slow potentials obtained by low pass filtering the raw signals. The activity was evoked by applying 6 μM NMDA and 8 μM 5-HT and the green bar indicates the duration of the light (60 s). (**B**) Bar plot showing the average locomotor-like frequency in wild type (n = 28, black circles), ChAT-Arch (n = 25, light

*Figure 3 continued on next page*

*Figure 3 continued*

green circles) and ChAT- eNpHR (n = 12, dark green circles) cords before the light (ANOVA, p=0.0533, two-stage linear step-up procedure, *: p=0.0141). (C–D) Time series of the change in absolute phase (C) and frequency (D) averaged for all experiments for the bilateral flexor (C1–D1) and ipsilateral flexor-extensor roots (C2–D2). (E) Averaged integrated ventral root discharge (Change in MN firing) for the ipsilateral flexor (E1) and extensor (E2) ventral roots. The statistics are obtained using a bootstrap t-test between ChAT- eNpHR (n = 12) and wild type cords (n = 28). The statistics are color-coded as indicated in the box below the records. The green rectangles indicate the duration of the light (60 s). (F–G–H) Bar plots showing the average change in the absolute phase (F), frequency of the bilateral flexors (G) for the 10 s just before and just after the light is turned on (Start Light, circles) and the 10 s just before and just after the light is turned off (After Light, squares) for wild type (black), ChAT-Arch (light green) and ChAT-eNpHR (dark green) animals. (H) Bar plots showing the average change of the ipsilateral flexor root for the 10 s just before and just after the light is turned on (Start Light, circles). Using a two-way ANOVA, we calculated the statistical differences between the three groups of animals (genetic Identity, shown above the bars) and the differences between light on and light off (light status; shown below the bars). The results of the ANOVA for the frequency changes were (Light status: $F_{(3, 248)}$ p<0.0001, Genetic identity $F_{(2,248)}$=0.0276, Interaction $F_{(6,248)}$: p<0.0001), for the absolute phase changes were (Light status: $F_{(3, 248)}$ p<0.0001, Genetic identity $F_{(2,248)}$ p<0.0001, Interaction $F_{(6,248)}$: p=0.8341), and for the changes in motoneuron firing were (Light status: $F_{(3, 248)}$ p<0.0001, Genetic identity $F_{(2,248)}$ p<0.0001, Interaction $F_{(6,248)}$: p<0.0001*p<0.05, **p<0.01, ***p<0.001, ****p<0.0001.

The following source data and figure supplement are available for figure 3:

**Source data 1.** Source data for Bar plots in *Figure 3B*.
**Source data 2.** Source data for Bar plots in *Figure 3F*.
**Source data 3.** Source data for Bar plots in *Figure 3G*.
**Source data 4.** Source data for Bar plots in *Figure 3H*.
**Figure supplement 1.** Effect of dopamine of the light-induced effects during drug-induced locomotor-like activity in ChAT-Arch animals.

from the control frequency before the light was turned on for both ChAT-Arch and ChAT-eNpHR cords (p<0.0001, p=0.0021; respectively, data not shown).

There was also a small but significant light-induced change in phase between the bilateral flexors in all cords (*Figure 3F*; WT: 13.3 ± 9.11°, p=0.0035, ChAT-Arch: 20.07 ± 12.47°, p<0.0001, ChAT-eNpHR: 19.33 ± 12.44°, p=0.0051). The light-induced phase changes were significantly greater in the Chat-Arch and ChAT-eNpHR cords compared to the wild type cords (ChAT-Arch: p=0.0008, ChAT-eNpHR: p=0.0167). The light-induced phase changes were larger in the ipsilateral flexor/extensor ventral roots (data not shown), and were greater in the ChAT-eNpHR cords than in the ChAT-Arch cords (ChAT-Arch: 46.87 ± 33.96°, ChAT-eNpHR: 74.79 ± 43.2°, p<0.0001). One possible explanation for the larger phase changes in the flexor/extensor roots and in the ChAT-eNpHR cords is that the extensor rhythmic drive is predominantly inhibitory whereas that to the flexors is mostly excitatory (*Endo and Kiehn, 2008*). Thus, under illumination, the membrane potential of extensor motoneurons may be hyperpolarized below the chloride equilibrium potential resulting in inhibitory drive potentials becoming depolarizing (See *Figure 3—figure supplement 1*). This may also explain the larger phase changes in the ChAT-eNpHR versus the ChAT-Arch cords because the chloride reversal potential is depolarized with the activation of the chloride pump (*Raimondo et al., 2012*).

## Intracellular recordings from flexor and extensor motoneurons reveal light-induced changes in rhythmic drive potentials

Our conclusions about the light-induced changes in the locomotor rhythm have been based on recordings of the rhythmic slow potentials extracted from the ventral root recordings. In many previous studies these have been shown to correspond to the timing of locomotor-drive potentials recorded intracellularly (*Kremer and Lev-Tov, 1997*; *Whelan et al., 2000*); nevertheless, we wanted to confirm the effects of the light on the rhythm with intracellular recordings. In the next set of experiments, we examined the effects of light on the firing, the intracellularly recorded rhythmic drive potentials and the intrinsic properties of individual flexor and extensor motoneurons in ChAT-Arch cords.

We first tested whether motoneuronal input resistance changed in green light under control conditions and in the drug-cocktail (*Figure 4—figure supplement 1A1*). We found that while there was a significant increase in input resistance after adding the drugs (control resistance: 44.8 ± 26.8 MΩ, NMDA-5-HT resistance: 76.38 ± 67.2 MΩ, p=0.0437), the resistance of the cells did not change in the light (p=0.1541, p=0.7171, respectively). The hyperpolarization induced by the light was greater in the presence of the drug cocktail (hyperpolarization control: 3.188 ± 1.5 mV, hyperpolarization NMDA-5-HT: 5.285 ± 3.8 mV, t-test p=0.0352, *Figure 4—figure supplement 1A2*) which was probably due to the increased input resistance in the drugs.

We recorded from 14 motoneurons (eight extensor, four flexor, two unidentified) in the L2 and the L5 segments. In the four flexor motoneurons, the light depressed the firing to 44.8% of the control value (measured over 60 seconds - pre-light spike number: 230 ± 102, light spike number: 103 ± 74). After the light, the flexor motoneuron firing increased by 9.1% (251 ± 122) compared to the pre-light control value. For the eight identified extensor motoneurons, the light decreased the firing of the motoneurons to 29.2% of the control firing level before the light was turned on (spike numbers: pre-light 342 ± 178, light 100 ± 120). Once the light was turned off the firing increased by 14.3% (391 ± 153) compared to the value before the light. The spike counts were made from trials in which the membrane potential ranged from −40.1 to −62.1 mV (mean value for 8 extensors −48 ± 4.6 mV; for 4 flexors −48.4 ± 5 mV). *Figure 4—figure supplement 2* illustrates motoneuron behavior during stable locomotor-like activity.

In both neurons illustrated in *Figure 4*, the intracellularly recorded rhythmic drive potentials disappeared or were greatly reduced in amplitude for 10–20 s after the light was turned on. When the flexor neuron (*Figure 4A1*), receiving primarily excitatory drive (*Endo and Kiehn, 2008*), was held at −50 mV, the light hyperpolarized the membrane potential to −55 mV and the rhythmic drive potential disappeared for approximately 10 s (*Figure 4A2*, red trace). The absence of synaptic drive was not due to a change in the amplitude of the locomotor drive potentials because when the cell was held at −61 mV, the amplitude of the drive potentials was similar to those recorded at −50 mV (*Figure 4A2*, green trace). These results strongly suggest that the rhythmic drive to the motoneuron has stopped. However, for the extensor motoneuron receiving predominantly inhibitory rhythmic drive (*Figure 4B1*), it is important to show that the decreased amplitude or the absence of drive potentials during hyperpolarization was not due to the membrane potential reaching the chloride equilibrium potential. This was not the case as shown in *Figure 4B2*. The effect of membrane hyperpolarization on the drive potentials is shown in the pre-light control portion of the records; from −48 to −68 mV, there is a reduction in the amplitude of the potentials without a change in frequency. However, during the light, the drive is absent or its frequency greatly reduced at all the holding potentials. This result confirms that this silencing was not due to the neuron reaching the chloride reversal potential when hyperpolarized, but rather to a loss of rhythmic synaptic inputs. In all recorded motoneurons, regardless of the nature of their synaptic inputs, the form and timing of the drive potentials were similar to the slow potentials recorded in the parent ventral root .

We were also concerned that the light-induced hyperpolarization might influence active membrane conductances that could contribute to the changes in the locomotor drive potentials. To show that the light-induced membrane potential change was not responsible for the decrease in frequency, we performed experiments in which the light was restricted to the caudal thoracic and rostral lumbar segments (*Figure 4—figure supplement 3*) so that the L5 segment was only minimally illuminated. In this configuration, motoneuron firing was barely affected in L5 (*Figure 4—figure supplement 3A and E*) compared to the substantial reduction that occurred during illumination of the whole cord (*Figure 4—figure supplement 3B and E*, dotted grey traces, p<0.0001 in L5). Nevertheless, the changes in frequency were similar in both L2 and L5, indicating that they were not due to changes in the behavior of the motoneurons themselves. However, the phase change in the flexor-extensor root pairs (*Figure 4—figure supplement 3C2*) was much less marked than when the entire lumbar cord was illuminated. As we discussed earlier, this is because the light can cause the drive potentials in L5 motoneurons to reverse if the hyperpolarization moves the membrane potential below the chloride equilibrium potential.

For both sets of experiments, the reduction in frequency was similar, which is perhaps surprising because the decrease in the firing of lumbar motoneurons as a population will be greatest when the whole lumbar cord is illuminated. However, because of the relatively low number of rostrally illuminated experiments (n = 7) and the high inter-animal variability, it is not possible to conclude

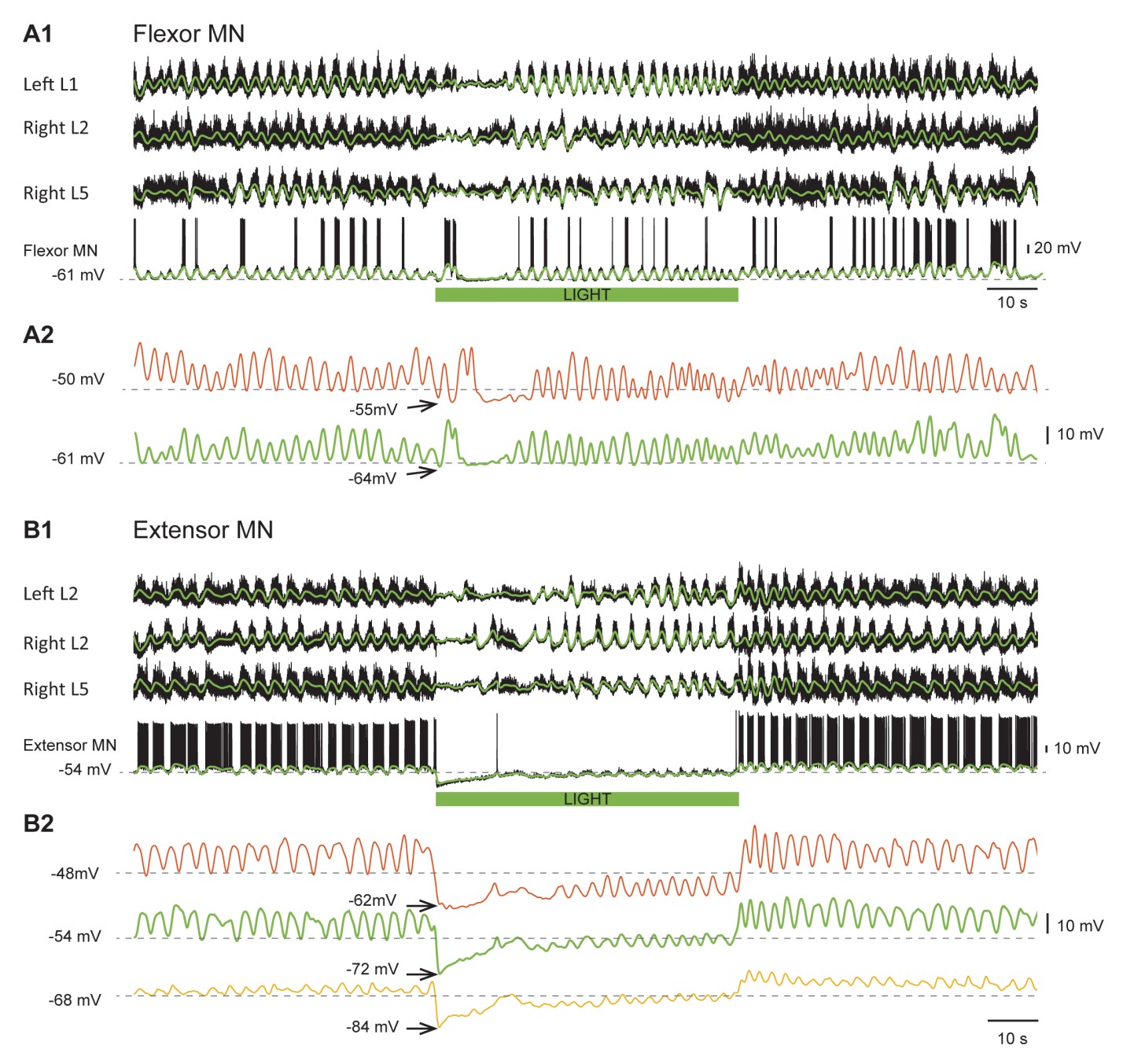

**Figure 4.** Light-induced hyperpolarization of cholinergic neurons affects the rhythmic synaptic drive to motoneurons. (**A1**) Locomotor-like activity recorded from the left L1 and the right L2 and L5 ventral roots together with a flexor motoneuron in the left L1 segment (black traces). The superimposed green traces are the slow potentials obtained by low pass filtering the raw signal. (**A2**) Low pass filtered records of the membrane potential of the same motoneuron held at two different membrane potentials (−50 mV - red trace; −61 mV - green trace) show that the locomotor drive potentials transiently disappeared for ~10 s after the light. (**B1**) Locomotor-like activity recorded from the left L2 and the right L2 and L5 ventral roots together with an extensor motoneuron in the right L5 segment (black traces). The superimposed green traces are the slow potentials obtained by low pass filtering the raw signals. (**B2**) Low pass filtered records of the membrane potential of the same motoneuron held at three different membrane potentials (−48 mV - red trace; −54 mV - green trace; −68 mV – yellow trace) show that the locomotor drive potentials transiently disappeared for 10–15 s after the light. The green bar below the intracellular recording indicates the duration of the light.

The following source data and figure supplements are available for figure 4:

**Figure supplement 1.** Properties of Motoneurons in ChAT-Arch mice.

*Figure 4 continued on next page*

*Figure 4 continued*

**Figure supplement 1—source data 1.** Input resistance MN ChAT-Arch spinal cords.

**Figure supplement 1—source data 2.** Intracellular hyperpolarization MN ChAT-Arch spinal cord.

**Figure supplement 2.** Green light suppresses the firing of individual flexor and extensor motoneurons in ChAT-Arch cords.

**Figure supplement 3.** The light-induced changes in the locomotor-like rhythm are not due to light-induced membrane potential changes on the intrinsic motoneuron properties in ChAT-Arch cords.

definitively that more limited illumination is as effective at reducing the locomotor frequency as the whole lumbar cord illumination. Nevertheless, we suggest that the effects of reduced motoneuron firing may be most prominent in the rostral lumbar segments because these have been shown to be the most rhythmogenic (*Cazalets et al., 1995*; *Kjaerulff and Kiehn, 1996*; *Kremer and Lev-Tov, 1997*).

## Experiments in animals expressing channelrhodopsin-2 in cholinergic neurons

The experiments with ChAT-Arch and ChAT-eNpHR revealed that reducing firing in cholinergic interneurons and motoneurons slowed the locomotor rhythm. In the next set of experiments, we investigated whether increasing the discharge of cholinergic neurons would accelerate the rhythm. To achieve this, we used animals in which channelrhodopsin-2 was expressed in cholinergic neurons. We examined the effects of exciting cholinergic neurons with a train of blue light pulses (100 Hz) on the locomotor-like activity induced by NMDA and 5-HT in ChAT-ChR2 cords and compared the effects of the same protocol applied to animals in which EGFP was expressed in ChAT$^+$ neurons. We used a train of pulses because continuous blue light damaged the cord, probably due to excessive heating. ChAT-EGFP animals were used as controls because EGFP is excited by blue light and could potentially have non-specific, light-dependent effects on the rhythm. However, the ChAT-EGFP cords showed minimal effects to blue light exposure (n = 8; *Figure 5—figure supplement 1*). By contrast, blue light produced a significant increase in motoneuron firing in the ChAT-ChR2 cords (n = 15; *Figure 5E*, *Figure 5—figure supplement 2*) that was greater in the extensor-dominant ventral roots than in the flexor-dominant roots (Extensor: 121 ± 60.29%; Flexor: 81.07 ± 34.89%, p=0.0002; 2-way ANOVA (see Materials and methods), light status: F(3,112) p<0.0001, root: F(1,112): p=0.0008, interaction F(3,112): p=0.0039). Concomitantly, we observed a significant increase in the frequency of the locomotor rhythm (*Figure 5D*; 2-way ANOVA, light status: F(2,63) p=0.0069, genetic identity F(1,63): p=0.0402, interaction F(2,63): p=0.0156). The maximum frequency occured at different times in individual experiments. When averaged across experiments, it increased by 13.34 ± 8.94%, in the ChAT-ChR2 cords, compared to 4.94 ± 4.43% (p=0.0006) in the ChAT-EGFP cords.

The light-induced increase in frequency had a slower time course than the decrease in frequency observed in the ChAT-Arch/eNpHR cords, with the maximum frequency for the averaged time series occurring at ~30 s after the light was turned on (*Figure 5D*). We also observed a small, but significant change in the phasing (*Figure 5C*) compared to the ChAT-EGFP controls, when motoneurons were depolarized at the beginning of the light for the bilateral flexors (ChAT-EGFP: 12.04 ± 5.38°, ChAT-ChR2: 27.38 ± 13.68°, p=0.0003; 2-way ANOVA, light status F (3,84): p=0.1943, genetic identity F(1,84): p=0.0004, interaction F(3,84): p=0.4827) and the ipsilateral flexor/extensors (ChAT-EGFP: 13.67 ± 5.94°, ChAT-ChR2: 24.34 ± 13.2°, p=0.0320, 2-way ANOVA, Light Status F(3,84): p=0.0233, genetic identity F(1,84): p<0.0001, interaction F(3,84): p=0.1084).

The locomotor frequency was slower in the ChAT-ChR2 spinal cords when compared to ChAT-EGFP cords (*Figure 5B*, frequency in ChAT-EGFP mice: 0.53 ± 0.07 Hz, frequency in ChAT-ChR2 mice: 0.42 ± 0.03 Hz, t-test p<0.0001).

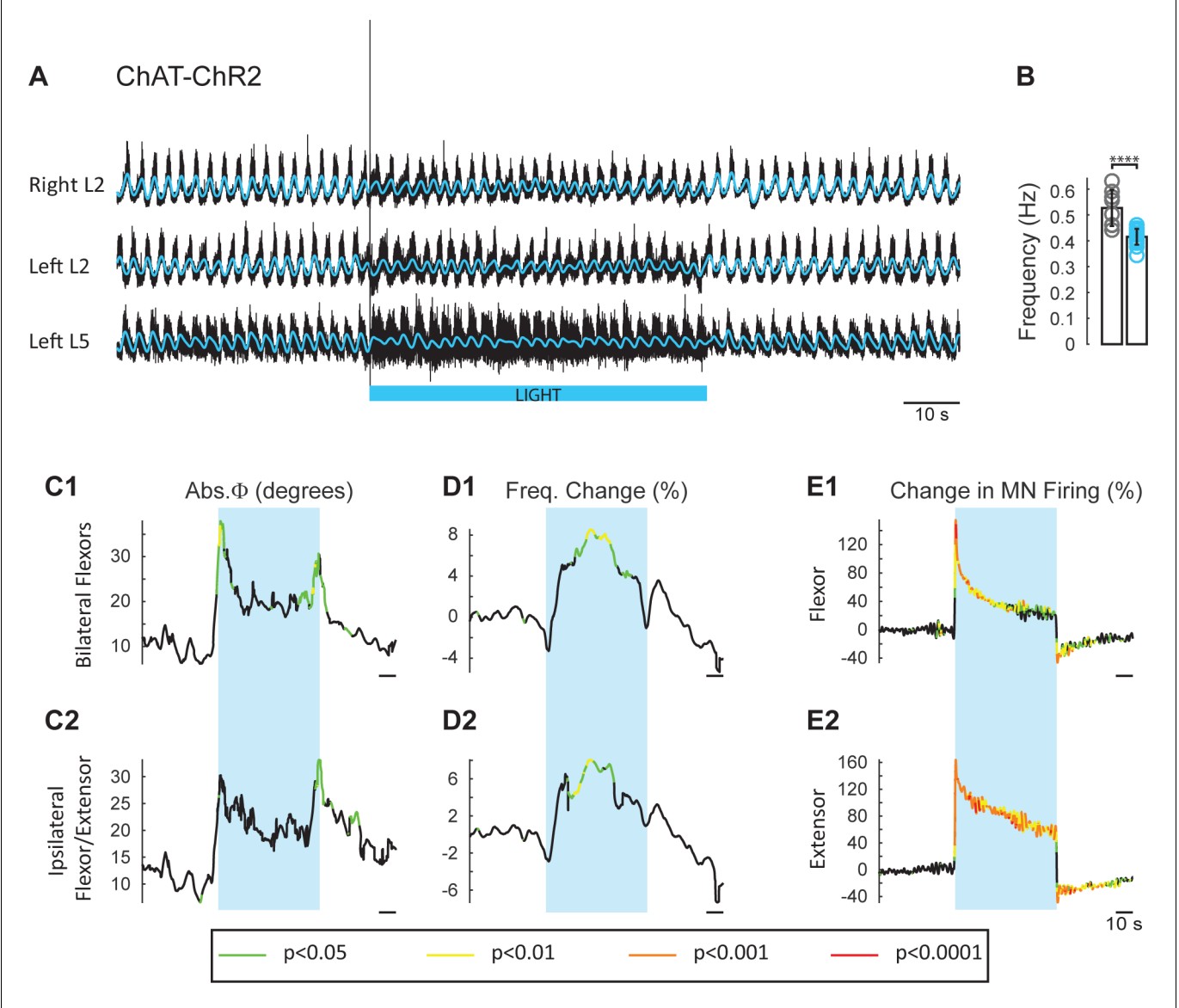

**Figure 5.** Light-induced depolarization of cholinergic neurons alters the frequency and phasing of drug-induced locomotor-like activity. (**A**) Locomotor-like activity recorded from the right L2 and the left L2 and L5 ventral roots (black traces) of a P1 isolated spinal cord from a ChAT-ChR2 mouse. The superimposed blue traces are the slow potentials obtained by low pass filtering the raw signal. Locomotor-like activity was evoked by applying 5 µM NMDA and 10 µM 5-HT. Activation of channelrhodopsin was achieved using a train of blue stimuli (1 ms at 100 Hz) for 60 s as indicated by the blue bar under the traces. (**B**) Bar plot showing the average locomotor-like frequency in Chat-EGFP (n = 8, grey circles) and ChAT- Channelrhodopsin (n = 15, blue circles) before the light. Note, that the frequency of the rhythm is significantly slower in the ChAT-ChR2 cords (t-test, ****p<0.0001). (**C–D**) Time course of the changes in phase (**C**) and frequency (**D**) averaged for all experiments for the bilateral flexors (**C1–D1**) and ipsilateral flexor-extensors (**C2–D2**). (**E**) Averaged integrated neurograms (Change in MN firing) for the ipsilateral flexor (**E1**) and extensor (**E2**) ventral roots. The statistics are obtained using a bootstrap t-test between ChAT-ChR2 and ChAT-EGFP cords and are color-coded as indicated in the box below the traces. The blue rectangles indicate the duration of the light (60 s).

The following source data and figure supplements are available for figure 5:

**Source data 1.** Source data for Bar plots in *Figure 5B*.

**Figure supplement 1.** Effect of blue light on drug-induced locomotor-like activity in ChAT-EGFP cords.

**Figure supplement 2.** Blue light increases the firing of motoneurons during locomotor-like activity in a ChAT-ChR2 cord.

## The light-induced disruption of locomotor-like activity in the ChAT-Arch animals persists in the presence of cholinergic blockade

It is not clear from these data, whether the effects of light on the locomotor-like rhythm are due to changes in the firing of motoneurons, of cholinergic interneurons, or of preganglionic neurons, all of which express the opsins when driven by the ChAT promoter. Our previous work showed that stimulation of motoneuron axons can evoke an episode of locomotor-like activity in the presence of cholinergic antagonists (*Mentis et al., 2005*). If the light-induced effects on the drug-induced rhythms engage a similar mechanism, then they should persist in the presence of the cholinergic antagonists. To test this hypothesis, we compared the effects of light in the ChAT-Arch spinal cords on the 5-HT-NMDA induced rhythm in the presence and absence of a cocktail of atropine, mecamylamine and dihydo-$\beta$-erythroid hydrobromide (dh$\beta$e). These antagonists degraded the locomotor-like rhythm (*Figure 6A*) under control conditions and this was reflected by a decrease in its frequency (*Figure 6B*) in both the wild type (n = 10, WT 0.45 ± 0.06 Hz, WT + antagonists: 0.34 ± 0.028 Hz, p<0.0001) and the ChAT-Arch cords (n = 6, ChAT-Arch: 0.53 ± 0.02 Hz, ChAT-Arch + antagonists: 0.39 ± 0.03 Hz, p<0.0001). In addition, there was also a change in phase between the bilateral flexors in the cholinergic blockers (*Figure 6C*, *Figure 6—figure supplement 1*, WT: 7.86 ± 2.59°, WT + antagonists: 27.96 ± 11.45°, p<0.0001; ChAT-Arch: 10.38 ± 5.66°, ChAT-Arch + antagonists: 23.74 ± 10.74°, p<0.0001).

In the presence of the cholinergic antagonists, the ChAT-Arch preparations showed a significant light-induced decrease in motoneuron firing compared to the wild type cords (WT: 6.36 ± 8.67%, WT + antagonists: 8.85 ± 7.39%, ChAT-Arch: −57.92 ± 12.76%, ChAT-Arch + antagonists: −59.12 ± 9.06%, p<0.0001, *Figure 6E*). The light-induced slowing of the locomotor-like rhythm persisted in the presence of the cholinergic antagonists (*Figure 6D*, Start Light). Under control conditions (no cholinergic antagonists) the light-induced change in frequency was −7.98 ± 3.9% and it increased to −12.75 ± 1.96% in the presence of the antagonists although not significantly (p=0.0914). Similarly, the light-induced phase change in the bilateral flexor motoneurons (WT: 11.5 ± 6.6°; ChAT-Arch: 15.46 ± 6°) significantly increased for both wild type and ChAT-Arch cords in the presence of the antagonists (WT: 29.76 ± 16.38°; ChAT-Arch: 47.25 ± 14.19°; p=0.0002; *Figure 6C*). The cholinergic antagonists did not block the rebound in frequency after the light was turned off (ChAT-Arch: 7.19 ± 5.61%, ChAT-Arch + antagonists: 12.7 ± 6.76%, p=0.0519). Collectively, these data indicate that the light-dependent effects are mediated by a non-cholinergic pathway, and are consistent with the earlier results showing that ventral root stimulation can activate the locomotor CPG in the presence of cholinergic antagonists (*Mentis et al., 2005*). They further suggest that cholinergic interneurons and preganglionic autonomic neurons are unlikely to be responsible for the light-induced slowing and the disruption of the rhythm.

## Light-induced hyperpolarization of *Isl1*-Arch cords slows the locomotor rhythm

To obtain additional evidence that the light-induced changes in frequency and phase were due to changes in the firing of motoneurons rather than some other neuronal class, we repeated the experiments in cords in which Arch was expressed in *Isl1*[+] neurons. *Isl1* is expressed in primary afferents, dI3 interneurons, motoneurons, and sympathetic motoneurons (*Figure 7A1-3*; *Tsuchida et al., 1994*; *Pfaff et al., 1996*; *Bui et al., 2013*; *Huber et al., 2013*). We hypothesized that if light-induced changes in motoneuron firing regulate the decreased locomotor frequency in the ChAT-Arch cords then we should observe similar effects in the *Isl1*-Arch cords. Consistent with this notion, light disturbed the locomotor rhythm and reduced its frequency (*Figure 7B,D*). In addition, there was a rebound increase in motoneuron firing (*Figure 7E*) and frequency (*Figure 7D*) when the light was turned off. The light-induced changes in frequency were not statistically different from those in the ChAT cords (Start light: −9.31 ± 13.92%, p=0.5474, End Light: 1.67 ± 13.83%, p=0.7287, After Light: 12.16 ± 9.5%, p=0.1434; 2-way ANOVA, Light status: p<0.0001, Genetic identity: p=0.2731, Interaction: p<0.0001; data not shown). Furthermore, in five additional *Isl1*-Arch cords, we removed the dorsal part of the cord over L1 to L5 (*Dyck and Gosgnach, 2009*; *Dougherty and Kiehn, 2010*; *Zhong et al., 2010*) to eliminate some of the *Isl1*[+] di3 neurons. We found that the effects of light on the locomotor rhythm were similar to those in the intact *Isl1*-Arch cords except that the decrease in frequency was maintained during the light (*Figure 7—figure supplement 1*).

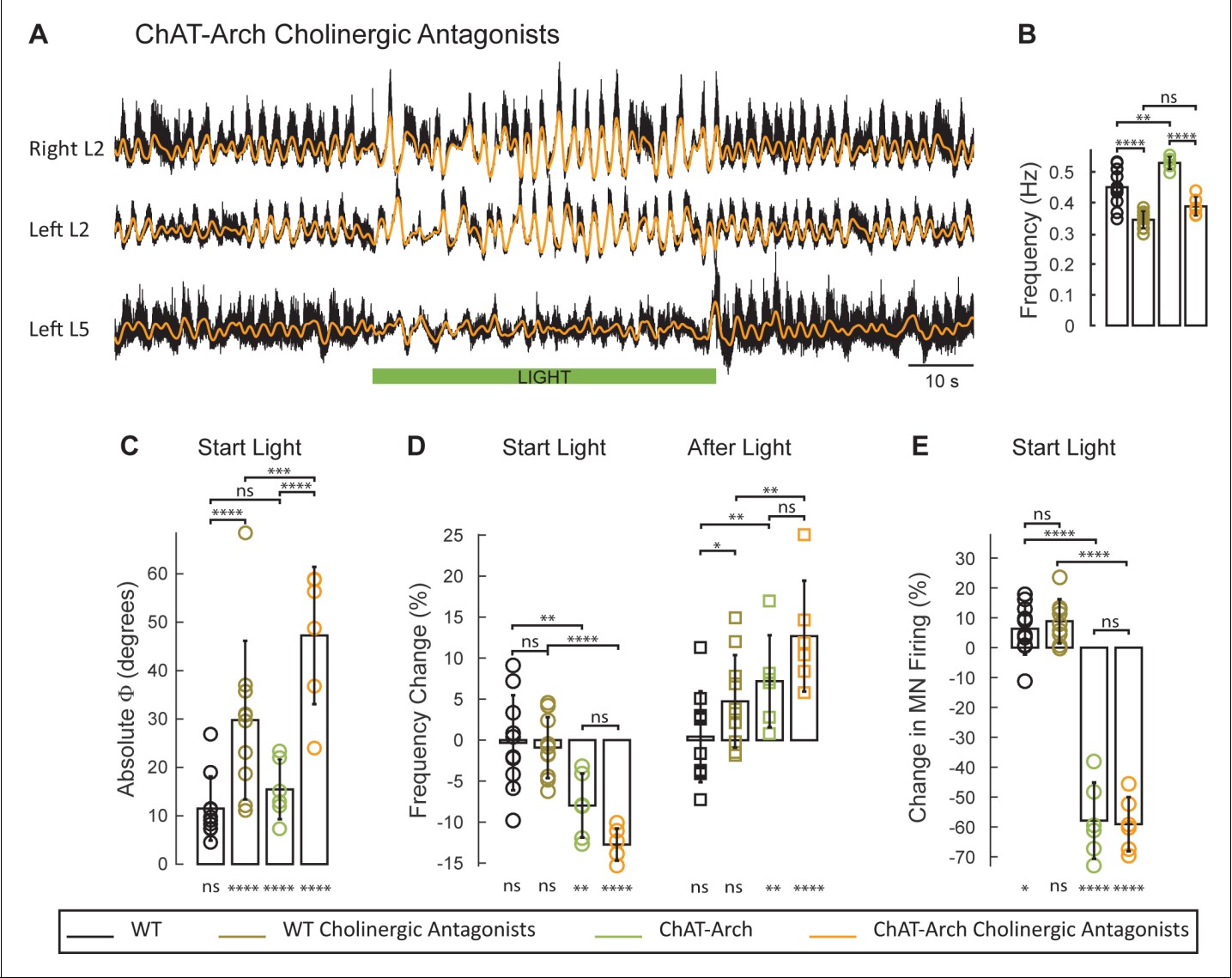

**Figure 6.** The light-induced disruption of the locomotor-like rhythm is exacerbated in the presence of cholinergic antagonists. (**A**) Ventral root recordings (Right L2, and Left L2 and L5) showing the effect of light on the locomotor-like rhythm evoked by 5 µM NMDA and 10 µM in the presence of 50 µM mecamylamine, 50 µM dh$\beta$E, and 5 µM atropine. The superimposed orange traces are the slow potentials obtained by low pass filtering the raw signal. The green bar indicates the duration of the light (60 s). (**B–E**) Comparison between WT (black, brown), ChAT-Arch (green, orange) cords in the absence or presence of cholinergic antagonists. (**B**) Bar plot showing the average locomotor-like frequency in WT (n = 10) and ChAT-Arch (n = 6) cords under control conditions (no light) in the absence or presence of cholinergic antagonists (ANOVA, p<0.0001). (**C–E**) Bar plots showing the average change in the absolute phase (**C**), frequency (**D**) of the bilateral flexors for the 10 s just before and just after the light is turned on (Start Light, circles) and the 10 s just before and just after the light is turned off (After Light, squares). (**E**) Bar plot showing the averaged integrated neurogram (Change in MN firing) of the ipsilateral flexor root for the 10 s just before and just after the light is turned on (Start Light, circles) for WT (black, brown), ChAT-Arch (green, orange) cords in the absence or presence of cholinergic antagonists. Using a two-way ANOVA we calculated the statistical differences between the three groups of animals (genetic Identity, shown above the bars) and the differences between light on and light off (light status, shown below the bars). The results of the ANOVA for the changes in the variables were: absolute phase (Light status: F (3,112) p<0.0001, Genetic identity/Drug treatment: F(3,112) p<0.0001, Interaction F(9,112) p=0.0017), frequency (Light status: F (3,112) p<0.0001, Genetic identity/Drug treatment: F(3,112) p=0.3073, Interaction F(9,112) p<0.0001), and motoneuron firing (Light status: F (3,112) p<0.0001, Genetic identity/Drug treatment: F(3,112) p<0.0001, Interaction F(9,112) p<0.0001). *p<0.05, **p<0.01, ***p<0.001, ****p<0.0001.

The following source data and figure supplement are available for figure 6:

**Source data 1.** Source data for Bar plots in *Figure 6B*.

**Source data 2.** Source data for Bar plots in *Figure 6C*.

*Figure 6 continued on next page*

*Figure 6 continued*

**Source data 3.** Source data for Bar plots in *Figure 6D*.

**Source data 4.** Source data for Bar plots in *Figure 6E*.

**Figure supplement 1.** Effect of green light during drug-induced locomotor-like in wild type cords in the presence of cholinergic blockers.

Collectively, this set of experiments further emphasizes that motoneurons are the most likely neurons responsible for the disruption of the rhythm in both the ChAT-Arch and *Isl1*-Arch cords.

## The light-induced changes in the locomotor-like rhythm in ChAT-Arch cords persist during partial blockade of gap junctions by carbenoxolone

In the next set of experiments, we investigated whether the light-induced effects might be mediated by gap junctions. In the adult zebrafish, motoneurons and V2a interneurons are connected by hybrid chemical/electrical synapses. Manipulating motoneuron membrane potential can influence firing of the V2a interneurons and alter the swimming rhythm (*Song et al., 2016*). In the larval *Drosophila*, it was shown recently that motoneurons also regulate the locomotor activity through gap junctions (*Matsunaga et al., 2017*). To establish whether gap junctions mediated the effects of light on the rhythm in the ChAT-Arch cords, we first determined the effect of bath-applied carbenoxolone (CBX, 100 µM) on the electrical connections between motoneurons (*Figure 8A–C*). Using whole-cell recordings from motoneurons, we recorded the antidromically-evoked short-latency depolarizations at intensities subthreshold for the antidromic action potential (*Figure 8A–B*, *Personius et al., 2007*) or during collision of orthodromic and antidromic action potentials (*Chang et al., 1999*; *Tresch and Kiehn, 2002*).The voltage-independence of the amplitude of these short latency depolarizations confirms that they are mediated by gap junction coupling (*Figure 8B*). Using the collision protocol, we compared the amplitude of the evoked short latency depolarizations, before and after applying carbenoxolone (*Tresch and Kiehn, 2000*). The first significant effect appeared after 10 min in the drug, when the amplitude of the depolarizations had decreased to 71% of the pre-drug level (n = 5, p<0.05). At 30 min in the drug, the amplitude was further reduced to 41% of the pre-drug level (p<0.001). The reduction in the amplitude of the short latency depolarizations was not due to a corresponding reduction in the membrane resistance of the motoneurons because the resistance was unchanged after 40 min in the drug (control 20 MΩ, 40 min 18.4 MΩ, p>0.9999, n = 5 motoneurons). Because the locomotor-like rhythm becomes highly unstable after 40 min in carbenoxolone, we examined the effects of light on the rhythm between 20 to 40 min after drug application. Assuming that any putative gap-junctions between motoneurons and interneurons will be depressed with a similar time course to the short-latency depolarizations in motoneurons  these experiments were performed with an incomplete block of gap junctional communication.

Under control conditions (no light), carbenoxolone decreased the frequency of the rhythm (*Figure 8E*) in both wild type (*Figure 8—figure supplement 1*, 0.45 ± 0.05 Hz, CBX: 0.29 ± 0.04 Hz, p=0.0003, n = 5) and ChAT-Arch cords (*Figure 8D*, 0.42 ± 0.07 Hz, CBX: 0.28 ± 0.04 Hz, p=0.0002, n = 6). In the ChAT-Arch cords, the light-induced depression of motoneuron firing was similar with or without carbenoxolone (*Figure 8H*; ChAT-Arch: −66.7 ± 9.6%, ChAT-Arch + CBX: −74 ± 17.9%, p=0.3102). By contrast, the light-induced change in phase was significantly greater in the presence of carbenoxolone (*Figure 8F*; ChAT-Arch: 32.88 ± 19.54°; ChAT-Arch + CBX: 55.77 ± 27.22°, p<0.0001). The decrease in frequency induced by the light still occurred but was not significantly smaller in the presence of carbenoxolone in the ChAT-Arch cords (*Figure 8G*; ChAT-Arch: −6.36 ± 1.8%; ChAT-Arch + CBX: −9.4 ± 3.4%, p=0.3215). After the light, the rebound in frequency was still observed but it was smaller than when the drug was not present (ChAT-Arch: 15.62 ± 12.21%, ChAT-Arch + CBX: 7.89 ± 3.95%, p=0.0145). These results suggest that carbenoxolone-sensitive gap junctions do not mediate the light-dependent effects.

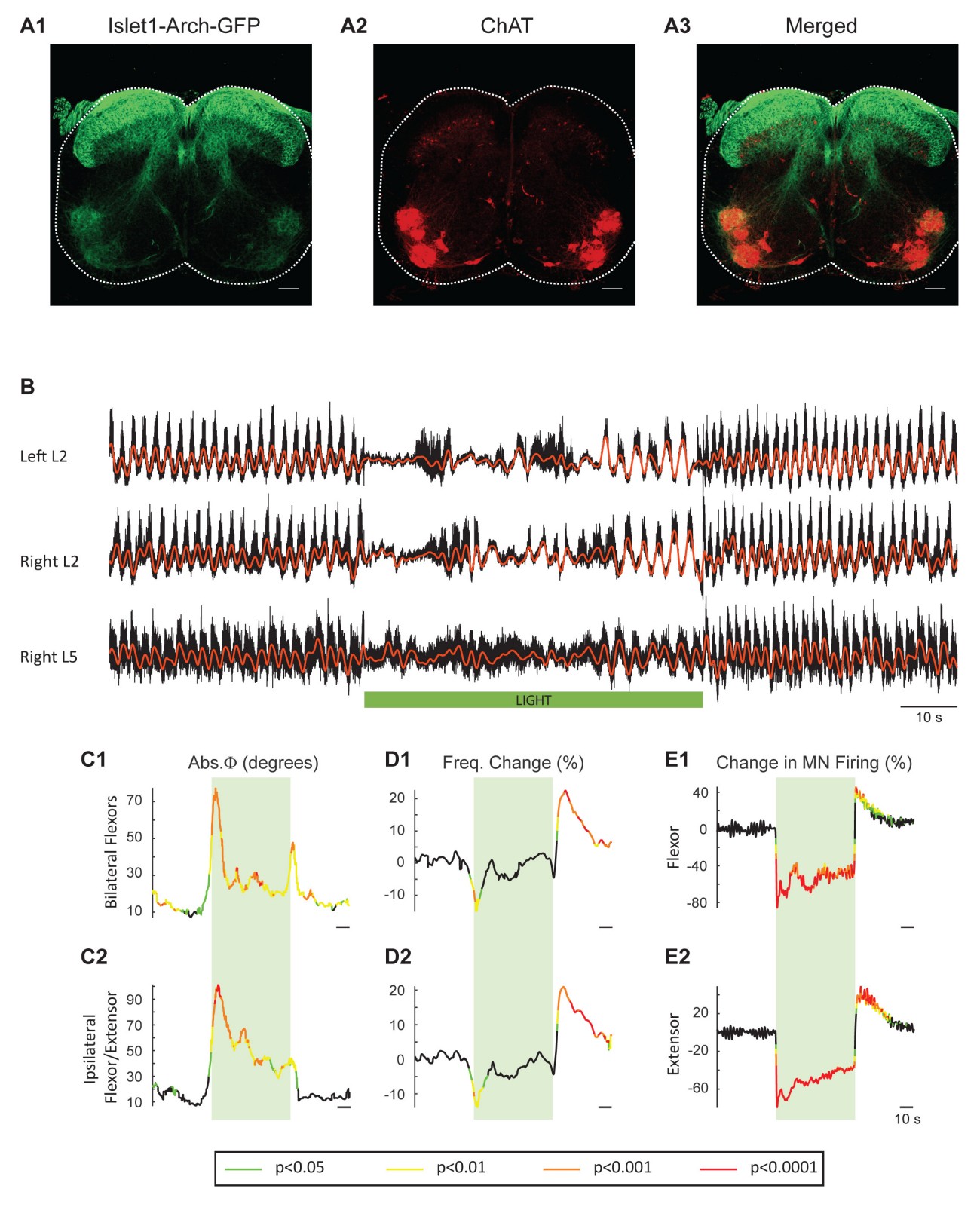

**Figure 7.** Light-induced hyperpolarization of *Isl1* neurons alters the frequency and phasing of locomotor-like activity. (**A**) Z-stack projection of 10X images (4 µm) of a 60 µm section of the L5 segment of a P2 *Isl1*-Arch mouse spinal cord showing archaerhodopsin (green, **A1 and A3**) and ChAT-positive (red, **A2 and A3**) neurons and the merged image (**A3**). The white scale bars represent 100 µm. (**B**) Locomotor-like activity recorded from the left L2 and the right L2 and L5 ventral roots (black traces) of a P1 isolated spinal cord from an *Isl1*-Arch mouse. The superimposed red traces are the slow

*Figure 7 continued*

potentials obtained by low pass filtering the raw signal. Locomotor-like activity was evoked by applying 5 µM NMDA and 10 µM 5-HT. The timing of the green light (60 s) is indicated by the green bar below the traces. (**C–D**) Time series of the change in absolute phase (**C**) and frequency (**D**) averaged for all experiments for the bilateral flexor (**C1–D1**) and ipsilateral flexor-extensor (**C2–D2**) roots. (**E**) Averaged integrated signals (Change in MN firing) for the ipsilateral flexor (**E1**) and extensor (**E2**) ventral roots. The statistics are obtained using a bootstrap t-test between *Isl1*-Arch (n = 8) and wild type cords (n = 28) and are color coded as indicated in the box below the records. The timing of the light (60 s) is indicated by the green rectangles.

The following figure supplement is available for figure 7:

**Figure supplement 1.** The light-induced perturbation of the locomotor-like rhythm is similar in dorsal-shaved and intact preparations of *Isl1*-Arch cords.

## In ChAT-Arch animals, the effects of light on the locomotor-like rhythm are attenuated in the presence of the AMPA-receptor antagonist NBQX

Motoneurons release glutamate or a glutamate-like neurotransmitter onto Renshaw cells (*Mentis et al., 2005*; *Nishimaru et al., 2005*) and activation of the locomotor CPG by ventral root stimulation is blocked by glutamate antagonists (*Mentis et al., 2005*). For this reason, we examined the effects of the AMPA-receptor antagonist NBQX on the rhythm. As shown previously (*Talpalar and Kiehn, 2010*), NBQX does not block the drug-induced locomotor-like rhythm but it does reduce its frequency. In agreement with this work, we found that NBQX decreased significantly the locomotor-like frequency in both wild type (*Figure 9—figure supplement 1*; WT: 0.44 ± 0.071, WT+ NBQX: 0.30 ± 0.049, p=0.0008, n = 6) and ChAT-Arch cords (*Figure 9A–B*; ChAT-Arch: 0.50 ± 0.082, ChAT-Arch + NBQX: 0.36 ± 0.03, p=0.0002, n = 7). When the light was turned on in the presence of NBQX, motoneuron firing transiently increased in the bilateral flexor ventral roots of the wild type cords (*Figure 9E*; WT: 2.6 ± 9.6%, WT+NBQX: 13.75 ± 11.9%, p=0.0213; *Figure 9—figure supplement 1*). However, in the ChAT-Arch cords the presence of NBQX depressed motoneuron firing further (ChAT-Arch: −61.54 ± 10.94%, ChAT-Arch + NBQX: −77.15 ± 8.58%, p<0.0001). While there was a small light-induced phase change in the wild type cords in the presence of NBQX, (*Figure 9C*; WT: 11.73 ± 3.46°, WT + NBQX: 28.19 ± 14.96°, p=0.0203), the light-induced changes were similar in the ChAT-Arch cords with and without NBQX (ChAT-Arch: 18.73 ± 5.587°, ChAT-Arch + NBQX: 26.39 ± 15.79°, p=0.2378). In wild type cords, light increased the locomotor frequency in the presence of NBQX (*Figure 9D*; WT: −2.11 ± 2.26% WT + NBQX: 6.07 ± 4.49%, p=0.0021) consistent with the increase in motoneuron firing (*Figure 9E*). By contrast, the light-induced decrease in the frequency of the Chat-Arch cords (p<0.001) was attenuated in the presence of NBQX (ChAT-Arch −6.41 ± 3.81%, ChAT-Arch + NBQX: −1.49 ± 5.49%, p=0.0418). After the light was turned off, the rebound in frequency was also blocked in the presence of NBQX (ChAT-Arch: 4.29 ± 5.6%, ChAT-Arch + NBQX: −1.49 ± 5.49%, p=0.0176). These results show that the AMPA-receptor blocker NBQX can substantially inhibit the light-induced effects on the locomotor-like rhythm in ChAT-Arch spinal cords and suggest that the effects are mediated, in part, by a glutamatergic mechanism.

## Discussion

We have shown that manipulating the firing of motoneurons during drug-induced locomotor-like activity is associated with changes in the frequency, phase and stability of the locomotor-like rhythm. Decreased motoneuron firing is accompanied by slowing of the rhythm whereas increased discharge is associated with an accelerated rhythm. Both effects were transient and decayed before the end of the 60 s light step or pulse train. In the ChAT-Arch cords there was also a transient increase in motoneuron firing and frequency once the light was turned off. Concomitant with the changes in frequency was a change in the absolute phase difference between the coupling of the bilateral flexor-dominated and the ipsilateral flexor/extensor ventral root slow potential activity. These results confirm that mammalian motoneurons in the neonatal period can access the locomotor rhythm generator and that motoneuron firing can regulate both the frequency and the phasing of locomotor-like activity.

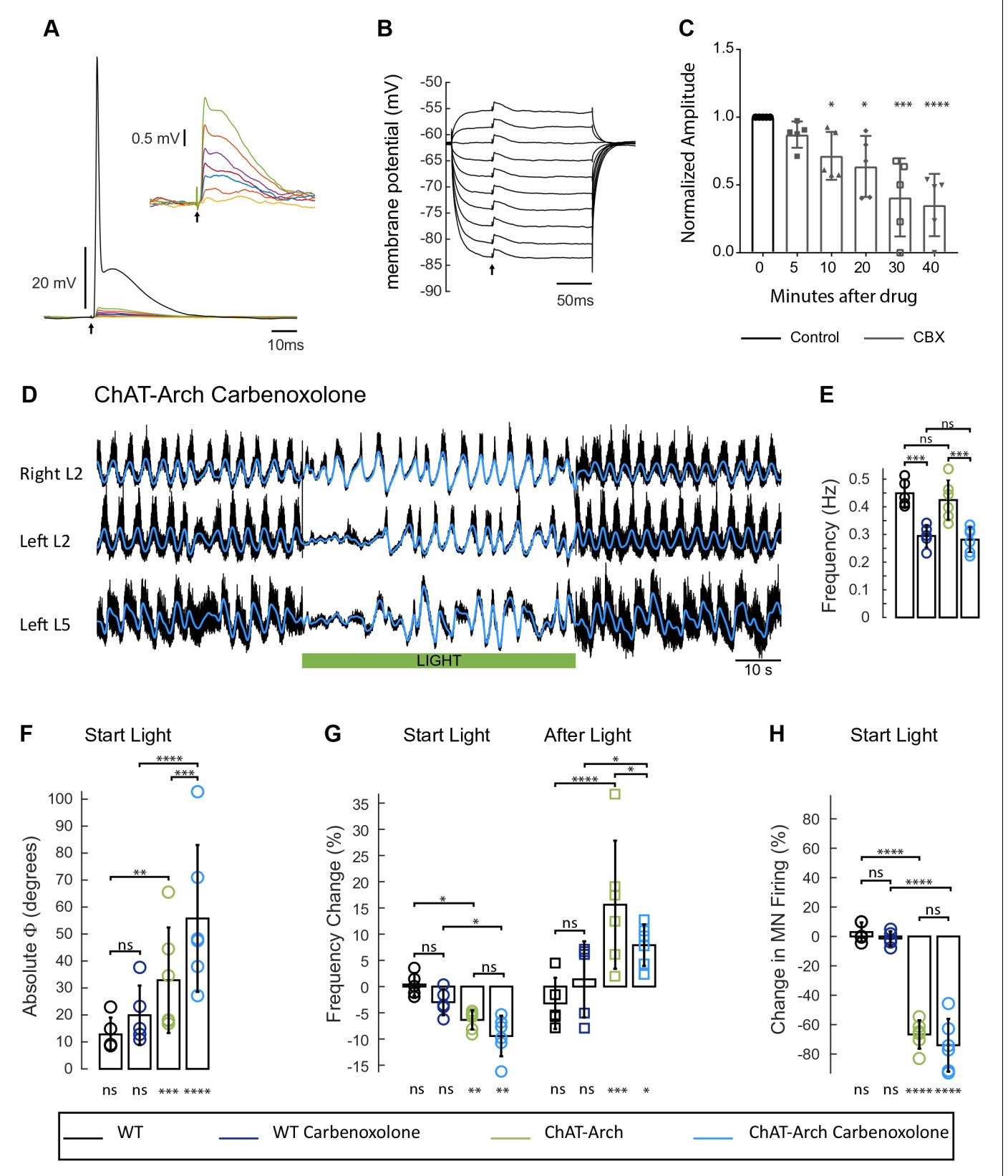

**Figure 8.** The light-induced disruption of the locomotor-like rhythm is maintained in the presence of the gap junction blocker carbenoxolone. (**A**) Superimposed intracellular responses of a motoneuron in a ChAT-Arch mouse to increasing intensity of antidromic stimulation (arrow) of the ventral

*Figure 8 continued on next page*

*Figure 8 continued*

root. Each trace is the average of 20 trials. The antidromic action potential is shown in black, the short-latency depolarizations (also shown in the inset) are colored and were evoked at stimulus intensities subthreshold for the action potential. (B) The amplitudes of the short-latency depolarizations are independent of membrane potential. The motoneuron was held at different membrane potentials by injecting hyperpolarizing or depolarizing current and a subthreshold stimulus was given every 500 ms (arrow). Each trace is an average of 20 trials. (C) Amplitude of the short latency depolarization in five motoneurons in control (before the drug) and 5, 10, 20, 30 and 40 min after applying 100 µM carbenoxolone. (D) Ventral root recordings (Right L2, and Left L2 and L5) showing the effect of light on the locomotor-like rhythm evoked by 5 µM NMDA and 10 µM 5-HT in the presence of 100 µM carbenoxolone. The superimposed blue traces are the slow potentials obtained by low pass filtering the raw signal. The green bar indicates the duration of the light (60 s). (E) Bar plot showing the average locomotor-like frequency in WT (n = 5) and ChAT-Arch (n = 6) cords under control conditions (no light) in the absence or presence of carbenoxolone. Note that in both the WT and the ChAT-Arch animals the frequency is significantly lower in the presence of carbenoxolone. ANOVA, p<0.0001. (F–H) Bar plots showing the average change in the absolute phase (F) and frequency of the bilateral flexors (G) for the 10 s just before and just after the light is turned on (Start Light, circles) and the 10 s just before and just after the light is turned off (After Light, squares). (H) Bar plot showing the averaged integrated neurogram (Change in MN firing) of the ipsilateral flexor root for the 10 s just before and just after the light is turned on for WT (black, dark blue), ChAT-Arch (light green, light blue) cords in the absence or presence of carbenoxolone. Using a two-way ANOVA, we calculated the statistical differences between the different groups of animals (genetic Identity, shown above the bars) and the differences between light on and light off (light status; shown below the bars). The results of the ANOVA for the changes in the phase were (light status: $F_{(3,72)}$ p<0.0001, Genetic identity/Drug treatment: $F_{(3,72)}$ p<0.0001, Interaction $F_{(9,72)}$ p=0.0379), for the frequency were: $F_{(3,72)}$ p<0.0001, Genetic identity/Drug treatment: $F_{(3,72)}$ p=0.004, Interaction $F_{(9,72)}$ p<0.0001) and for the change in motoneuron firing were: (Light status: $F_{(3,72)}$ p<0.0001, Genetic identity/Drug treatment: $F_{(3,72)}$ p<0.0001, Interaction $F_{(9,72)}$ p<0.0001). *p<0.05, **p<0.01, ***p<0.001, ****p<0.0001.

The following source data and figure supplement are available for figure 8:

**Source data 1.** Source data for Bar plots in *Figure 8C*.
**Source data 2.** Source data for Bar plots in *Figure 8E*.
**Source data 3.** Source data for Bar plots in *Figure 8F*.
**Source data 4.** Source data for Bar plots in *Figure 8G*.
**Source data 5.** Source data for Bar plots in *Figure 8H*.
**Figure supplement 1.** Effect of green light on drug-induced locomotor-like activity in wild type cords in the presence of carbenoxolone to block gap junctions.

## Contribution of preganglionic autonomic neurons and spinal interneurons to the light-induced changes in the frequency of the locomotor-like rhythm

One of the major limitations of the present work is that there are no specific markers for motoneurons. We chose to study ChAT and *Isl1* animals because motoneurons express both markers. However, cholinergic autonomic preganglionic neurons also express *Isl1* (*Tsuchida et al., 1994*) so that the effects of light could be attributable to silencing of their firing in addition to, or instead of, the silencing of motoneurons. Preganglionic neurons in the inter-mediolateral nucleus can exhibit oscillatory activity in spinal cord slices (*Logan et al., 1996*; *Spanswick et al., 1996*) and NMDA can induce coupled somatic motor and sympathetic rhythmic activity in an arterially perfused spinal cord preparation (*Chizh et al., 1998*). Furthermore, in the spinal cat, rhythmic activity can be recorded from sympathetic nerves that is coupled with the activity of hindlimb muscle nerves in response to L-DOPA administration (*Schomburg et al., 2003*). Little is known about the mechanisms responsible for this coupling, but it has been suggested to be due to a common input to sympathetic and somatic motoneurons from the locomotor central pattern generator (*Chizh et al., 1998*; *Deuchars, 2007*). *Chizh et al. (1998)* showed that synchronized rhythmic activity recorded from sympathetic postganglionic neurons was abolished by the nicotinic cholinergic antagonist hexamethonium consistent with their cholinergic phenotype. Our observation that the optogenetically-induced changes in frequency in the ChAT-Arch animals was preserved in the presence of nicotinic and muscarinic antagonists suggests therefore that the preganglionic sympathetic neurons are not responsible for the effects of light on the locomotor rhythm in the Chat-Arch animals. This result is consistent

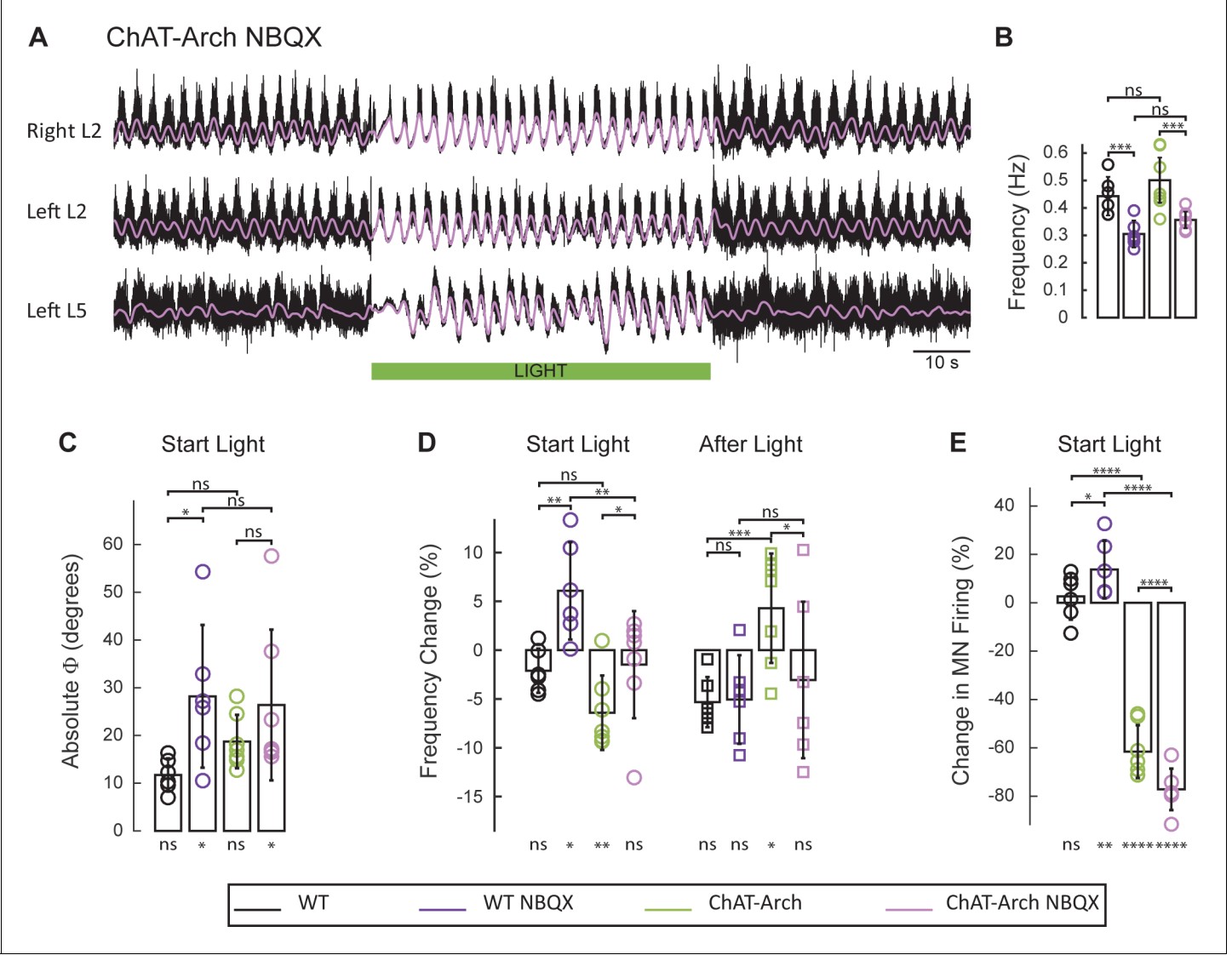

**Figure 9.** The light-induced disruption of the locomotor-like rhythm is attenuated in the presence of the AMPA-receptor antagonist NBQX. (**A**) Ventral root recordings (Right L2, and Left L2 and L5) showing the effect of green light on the locomotor-like rhythm evoked by 5 µM NMDA and 10 µM 5-HT in the presence of 10 µM NBQX. The superimposed light purple traces are the slow potentials obtained by low pass filtering the raw signal. The green bar indicates the duration of the light. (**B**) Bar plot showing the average locomotor-like frequency in WT (n = 6) and ChAT-Arch (n = 7) cords under control conditions (no light) in the absence or presence of NBQX. For both WT and ChAT-Arch animals the frequency of the rhythm is significantly slower in the presence of NBQX. ANOVA, p<0.0001 (**C–E**) Bar plots showing the average change in the absolute phase (**C**) and frequency (**D**) of the bilateral flexors for the 10 s just before and just after the light is turned on (Start Light, circles) and the 10 s just before and just after the light is turned off (After Light, squares). (**E**) Bar plot showing the averaged integrated neurogram (Change in MN firing) of the ipsilateral flexor root for the 10 s just before and just after the light is turned on for WT (black, dark purple), ChAT-Arch (green, light purple) cords in the absence or presence of NBQX. Using a two-way ANOVA, we calculated the statistical differences between the different groups of animals (genetic identity, shown above the bars) and the differences between light on and light off (light status; shown below the bars). (**B–C–D–E**) Comparison between WT (black, dark purple), ChAT-Arch (green, light purple) cords in the absence or presence of NBQX. The results of the ANOVA for the changes in the phase were (Light status: F (3,88) p=0.2436, genetic identity/Drug treatment: F(3,88) p<0.0001, interaction F(9,88) p=0.3893), for the frequency were (Light status: F (3,88) p=0.0630, Genetic identity/Drug treatment: F(3,88) p=0.1335, interaction F(9,88) p<0.0001) and for the motoneuron firing were (Light status: F (3,88) p<0.0001, Genetic identity/Drug treatment: F(3,88) p=0.0001, interaction F(9,88) p<0.0001). *p<0.05, **p<0.01, ***p<0.001, ****p<0.0001.

The following source data and figure supplement are available for figure 9:

**Source data 1.** Source data for Bar plots in *Figure 9B*.

**Source data 2.** Source data for Bar plots in *Figure 9C*.

*Figure 9 continued on next page*

*Figure 9 continued*

**Source data 3.** Source data for Bar plots in *Figure 9D*.

**Source data 4.** Source data for Bar plots in *Figure 9E*.

**Figure supplement 1.** Effect of green light on drug-induced locomotor-like activity in wild type cords in the presence of NBQX.

with our earlier finding that activation of the locomotor-like rhythm by stimulation of motor axons in the sciatic nerve (*Mentis et al., 2005*), which engages neither preganglionic nor cholinergic spinal neurons, persists in the presence of cholinergic antagonists.

In the ChAT-Arch cords, light will also affect the firing of a recently identified population of V0-derived interneurons that express the transcription factor Pitx-2 (*Zagoraiou et al., 2009*). These neurons fire either tonically or rhythmically during drug-induced locomotor-like activity and depolarize motoneurons by blocking their calcium-dependent potassium channels and reducing their after-hyperpolarization (*Miles et al., 2007*). However, normally phased locomotor activity was observed when the gene for ChAT was eliminated in these neurons (*Zagoraiou et al., 2009*) indicating that hyperpolarizing these neurons is unlikely to be responsible for the light-dependent changes in the locomotor rhythm. Furthermore, they do not appear to co-release glutamate because neither VGluT1 nor VGluT2 were expressed in the ChAT-deleted neurons (*Zagoraiou et al., 2009*).

A potential confound to the interpretation of our experiments arises because some cholinergic interneurons may also release glutamate in addition to acetylcholine. In the developing *Xenopus* tadpole, *Li et al. (2004)* described a class of caudally projecting interneurons that co-release glutamate and acetylcholine at their monosynaptic projections to motoneurons and premotor interneurons. Similarly, *Liu et al. (2009)* identified cholinergic terminals that co-expressed VGluT2 in Lamina VII of the ventral horn of adult rats, although it was not established where the cell bodies of the parent neurons were located. Moreover, no evidence was provided, in either case, that the co-release of glutamate by the cholinergic neurons had any influence on the locomotor CPG. Nevertheless, our data do not allow us to exclude some contribution from cholinergic interneurons that co-release glutamate.

In addition to motoneurons and preganglionic autonomic neurons, *Isl1* is expressed by primary afferents and glutamatergic di3 interneurons (*Bui et al., 2013*, *2016*). Hyperpolarizing primary afferents is unlikely to modify the locomotor rhythm because they are cut and are presumably inactive. The glutamatergic di3 neurons project monosynaptically to motoneurons and receive monosynaptic inputs from VGluT1[+] afferents. Furthermore, although the di3 neurons are in the pathway from afferents to the locomotor CPG, eliminating their VGluT2 expression resulted only in subtle alterations in over-ground locomotion (*Bui et al., 2013*). As a result, suppressing their firing by light should not account for the light-induced changes in locomotor function that we observed. Consistent with this hypothesis, when we removed the dorsal part of the spinal cord containing some of the *Isl1*[+] interneurons in 5 *Isl1*-Arch preparations, we found that the light depressed the frequency of the locomotor rhythm by a similar amount as in the intact spinal cord (*Figure 7—figure supplement 1*).

## Mechanism of the light-induced changes in the locomotor-like rhythm and the post-light rebound

The simplest hypothesis to account for the actions of motoneurons on the locomotor rhythm is that they are mediated through motoneuronal projections to Renshaw cells, by exerting disinhibitory effects on 1a inhibitory interneurons or ventral spino-cerebellar cells (*Hultborn et al., 1971*; *Lundberg and Weight, 1971*; *Jankowska et al., 1975*). However, several lines of evidence argue against the participation of this pathway in the excitatory effects of motoneurons on the locomotor CPG. First, the effects of motoneuron activity are enhanced and not blocked in the presence of muscarinic and nicotinic antagonists. Second, the excitatory effects of ventral root stimulation persist in the presence of the inhibitory antagonists bicuculline and strychnine. *Bonnot et al. (2009)* and *Machacek and Hochman (2006)* showed that ventral root stimulation could entrain disinhibited bursting in the neonatal mouse and rat spinal cords. In the mouse, the effect persisted in cholinergic

antagonists and was enhanced by the AMPA desensitization blocker cyclothiazide, suggesting a glutamatergic mechanism. Finally, ventral root stimulation can activate the locomotor rhythm when V1 neurons are hyperpolarized optogenetically (Falgairolle and O'Donovan, unpublished results).

An alternative possibility is that motoneurons synapse with an undiscovered class of excitatory interneuron that has access to the locomotor CPG. *Machacek and Hochman (2006)* reported that noradrenaline unmasks a recurrent excitatory pathway from motoneurons and attributed this to the recurrent activation of excitatory interneurons. Consistent with this idea, earlier reports described antidromic activation of glutamatergic EPSPs in motoneurons of neonatal rats (*Jiang et al., 1991*; *Ichinose and Miyata, 1998*). In the Ichinose and Miyata study, the latency of EPSPs was slightly but not significantly longer (6.9 ms vs 5.8 ms) than that of the disynaptic IPSP (assumed to be from Renshaw cells) suggesting that the excitatory EPSP probably originated disynaptically. If these putative excitatory interneurons also project to the locomotor CPG it could provide a mechanism linking motoneuronal firing to the CPG circuitry. Alternatively, it is possible that motoneurons release glutamate or another excitatory amino acid from their soma or dendrites that could excite spinal networks non-specifically. Calcium-dependent vesicular dendritic release of glutamate has been described for cerebellar Purkinje cells (*Duguid and Smart, 2004*; *Duguid et al., 2007*; *Shin et al., 2008*) and could, in principle, mediate the excitatory effects of motoneurons on the rhythm. A glutamatergic effect is consistent with the ability of the AMPA-receptor antagonist NBQX to partially block the effects of motoneuron firing on the rhythm and to abrogate activation of the locomotor CPG by motor axon stimulation. Because glutamate is essential for locomotor-like activity, we could not test the effect of blocking both NMDA and AMPA receptors on the light-dependent effects. *Caldeira et al. (2017)* recently showed that knocking out VGluT2 in ChAT$^+$ neurons did not alter the locomotor-like rhythm. This raises the possibility that the effects seen in our study were not mediated by motoneurons co-releasing acetylcholine and glutamate, but rather through a glutamatergic effect on the network itself. Alternatively, motoneurons might release glutamate or another excitatory amino acid (*Richards et al., 2014*) through a non-VGluT2 mechanism.

In ChAT-Arch, ChAT-eNpHR, and *Isl1*-Arch animals, we observed a transient increase in the firing of motoneurons and in the locomotor frequency when the light was turned off. This long-lasting effect (more than 10 s) could be due to intrinsic properties of motoneurons, a network effect or both. One possibility is that the recovery of the locomotor frequency that occurs during the light reflects a homeostatic compensation that is mediated by an increase in the excitatory interneuronal input to the CPG. When the light is turned off, the motoneuronal input is restored and together with the increased interneuronal input results in a transient increase in the locomotor frequency. Interestingly, NBQX blocked the rebound increase in frequency but not the rebound increase in motoneuron firing consistent with the idea that it uncouples the effects of motoneuron firing from the locomotor frequency.

## Comparison with other species

For many years, it was assumed that motoneurons transmitted the output of the vertebrate locomotor central pattern generator to muscles, without directly affecting the generator itself. Work in the *Xenopus* tadpole, however, revealed that rhythmically active premotor neurons, presumed to be part of the swimming CPG, received rhythmic cholinergic inputs that the authors argued originated from motoneurons (*Perrins and Roberts, 1995*). However, this conclusion is less certain now that it has been shown that glutamatergic spinal neurons also release acetylcholine (*Li et al., 2004*). Work in developing chick (*O'Donovan et al., 1994*; *Wenner and O'Donovan, 2001*) and mouse (*Hanson and Landmesser, 2003*) embryos established an important function for motoneuronal activity in the genesis of spontaneously occurring episodes of rhythmic motor activity. Calcium imaging of spinal neurons showed that fluorescent activity at the beginning of a spontaneous episode occurred within the lateral motor column and then spread dorso-medially and mediolaterally to encompass the whole cord (*O'Donovan et al., 1994*). Later experiments revealed that the initial activity in motoneurons is communicated to the rest of the network via the avian equivalent of Renshaw cells – R-interneurons (*Wenner and O'Donovan, 2001*).

Experiments in the adult zebrafish have shown that activity in motoneurons can modulate the activity of excitatory premotor excitatory V2a interneurons through hybrid chemical and electrical synapses (*Song et al., 2016*). Membrane potential changes in motoneurons can therefore influence the firing and transmitter release from the V2a neurons that have been implicated in

rhythmogenesis. Recently, it was also shown that in the *Drosophila larvae*, the activity of motoneurons regulates the frequency of the peristaltic crawling via a mechanism that depends on electrical coupling (*Matsunaga et al., 2017*). However, in the mammalian cord, electrical coupling between limb-moving motoneurons and interneurons has not been reported. Although mixed electrical/chemical synapses have been described in the adult rat cord (*Rash et al., 1996*), such hybrid synapses expressing connexin-36 do not appear in the rat or mouse cord until after the second or third neonatal week (*Bautista et al., 2014*), suggesting that they are probably not mediating the excitatory effects of motoneurons on the rhythm. Consistent with this idea, we found that the effects of hyperpolarizing ChAT$^+$ neurons on the locomotor rhythm were maintained in the presence of carbenoxolone even though gap-junctional coupling was only partially blocked (*Figure 8*).

## Function of motoneuron feedback to the CPG

The regulation of rhythmogenesis by motoneuronal discharge raises the question of its function during normal locomotion. It is difficult to generalize about the function of motoneuronal feedback onto the locomotor CPG because our experiments were performed in the isolated spinal cord in the absence of the descending and afferent inputs that normally accompany locomotion. One possibility is that feedback from motoneurons constitutes an efference copy of the motoneuron output that could be compared to ongoing motor commands and relayed to higher centers including the cerebellum. For a discussion of internal models of the motor system see *Wolpert and Miall (1996)*. A second possibility is that the output of motoneurons provides excitatory drive to the CPG that supplements the drive derived from interneurons. In some mice, the primary effect of reducing motoneuron discharge was a slowing of the rhythm (*Figure 2* and *Figure 4—figure supplement 2*: motoneurons 3 and 5) whereas in others the rhythm was transiently interrupted as shown by the lack of rhythmic slow potentials in the ventral root recordings (*Figure 4B1*) or by the absence of rhythmic drive potentials in the intracellular recordings (*Figure 4B2*; *Figure 4—figure supplement 2*: motoneurons 4, and 7). Under this latter condition, we suggest that the sudden removal of the excitatory input from motoneurons led to a brief cessation of the output of the CPG to motoneurons. The effect was transient either because of the recovery of motoneuron firing or because the reduced excitatory input to the CPG from motoneurons was rapidly compensated by excitatory inputs from other sources. We are not suggesting that the excitatory input from motoneurons is essential for the operation of the CPG but rather that it may be important in regulating the excitability of the CPG.

The excitatory effects of motoneuron activity onto the locomotor CPG represent a form of positive feedback because increasing motoneuron activity will accelerate the locomotor rhythm which in turn will increase motoneuron activity. Such a system is potentially unstable and we hypothesize that the excitatory motoneuronal input to the CPG is opposed by inhibitory input originating from the motoneuronal projections to Renshaw cells. According to this model, the feedback control of the CPG by motoneuron activity is regulated by the balance between recurrent excitation and recurrent inhibition. The ability of recurrent inhibition to inhibit the locomotor rhythm was demonstrated in the VGluT2 knockout mouse where ventral root stimulation terminated rhythmic activity by a cholinergic mechanism (*Talpalar et al., 2011*). This hypothesis may help to explain why the excitatory effects of motoneurons onto the CPG have been difficult to study systematically (*Bonnot et al., 2009*; *Machacek and Hochman, 2006*; *Humphreys and Whelan, 2012*) and why there is considerable inter-animal variability in the excitatory effects of motoneurons. Furthermore, this hypothesis provides a potential mechanistic explanation for the corelease of glutamate and acetylcholine from motoneurons whereby the excitatory input to the CPG is regulated by glutamatergic motoneuronal function and the inhibitory pathway by cholinergic function.

# Materials and methods

## Animals and dissection

Experiments were performed on Swiss Webster wild type (Taconic Laboratory) transgenic mice between the day of birth to post-natal day 3 (P0-P3). We used the following transgenic lines obtained from Jackson Laboratories: floxed archaerhodopsin-GFP (Arch, B6;129S-*Gt(ROSA) 26Sor*$^{tm35.1(CAG-aop3/GFP)Hze}$/J, stock# 012735), floxed archaerhodopsin-T-EGFP (ArchT, B6.Cg-*Gt (ROSA)26Sor*$^{tm40.1(CAG-aop3/EGFP)Hze}$/J, stock #021188), floxed halorhodopsin-EYFP (eNpHR, B6;129S-

$Gt(ROSA)26Sor^{tm39(CAG-hop/EYFP)Hze}$/J, stock# 014539), floxed channelrhodopsin-EYFP (ChR2, B6;129S-$Gt(ROSA)26Sor^{tm32(CAG-COP4*H134R/EYFP)Hze}$/J, stock# 012569), and floxed EGFP (EGFP, $Gt(ROSA)26Sor^{tm1.1(CAG-EGFP)Fsh}$/Mmjax, MMRRC stock# 32037-JAX) . We crossed these mice with a ChAT-Cre (B6;129S6-$Chat^{(tm1cre)Lowl}$/J, stock# 006410) line to obtain animals in which the optogenetic proteins were expressed in cholinergic neurons. In addition, we crossed the floxed archaerhodopsin lines with $Isl1$-Cre mice ($Isl1^{tm1(cre)Sev}$/J, stock# 024242). We have combined the results of the Arch-and ArchT animals, and we refer to them in the text as Arch.

Mouse pups were decapitated and then eviscerated. The preparations were then placed in a dissection chamber and continuously superfused with artificial cerebrospinal fluid (ACSF; concentrations in mM: 128 NaCl, 4 KCl, 1.5 $CaCl_2$, 1 $MgSO_4$, 0.5 $NaH_2PO_4$, 21 $NaHCO_3$, 30 D-glucose) bubbled with 95% $O_2$–5% $CO_2$. A ventral laminectomy was performed and the spinal cord together with the dorsal and ventral roots were removed from the body and placed in a recording chamber superfused with oxygenated ACSF at room temperature.

To remove the dorsal part of the spinal cord (*Figure 7—figure supplement 1*), the lumbar dorsal pia was removed. The spinal cord was then pinned dorsal up onto a strip of beeswax mounted on a small piece of plexiglass. The preparation was then transferred to the vibratome chamber containing a chilled oxygenated K-gluconate based solution (concentrations in mM: 130 K-Gluconate, 15 KCl, 0.05 EGTA, 20 Hepes, 25 D-glucose, 1 mM Kynurenic acid, 2 mM Na-pyruvate, adjusted to pH 7.4 with KOH). The vibratome was then used to remove the dorsal part of the lumbar spinal cord (at least 250 µM, or until the central canal was visible). The cord was then transferred to a recording chamber at room temperature and allowed to recover for at least 30 min. All experiments were carried out in compliance with the National Institutes of Neurological Disorders and Stroke Animal Care and Use Committee (Animal Protocol Number 1267–12 and 1267–15).

Locomotor-like activity was elicited with N-Methyl-D-Aspartic acid (NMDA, 5–6 µM) and serotonin creatinine sulfate monohydrate (5-HT, 8–10 µM). In one set of experiments (*Figure 3*, *Figure 3—figure supplement 1*), we used NMDA (5 µM), 5-HT (10 µM), and dopamine hydrochloride (DA, 50 µM). Cholinergic receptors were blocked by using 5 µM atropine methyl bromide salt, 50 µM mecamylamine hydrochloride and 50 µM dihydo-$\beta$-erythroid hydrobromide (dh$\beta$e). We used carbenoxolone disodium salt (CBX, 100 µM) as a non-specific, broad-spectrum blocker of gap junctions, and NBQX disodium salt (5–10 µM) to block AMPA receptors. NMDA, 5-HT, Dopamine, mecamylamine, atropine, and carbenoxolone were obtained from Sigma-Aldrich (St. Louis, MO), while dh$\beta$e and NBQX were obtained from Tocris (Minneapolis, MN).

## Activation of opsins

The protocol for activating the opsins was a control period of 60 s in the absence of light, 60 s with the light turned on and a recovery period of 60 s. Spinal cords expressing archaerhodopsin or halorhodopsin were stimulated with continuous green light (540–600 nm) applied from a 3 mm diameter light guide connected to a light emitting diode (X-Cite, Excelitas Technologies) positioned over the ventral surface of the cord to illuminate all the lumbar segments. In one set of experiments (*Figure 4—figure supplement 3*), we applied the green light to the rostral lumbar and caudal thoracic segments (leaving the L5 segment unilluminated). In another set of experiments (*Figure 2—figure supplement 2*), the light intensity was varied and the cord was illuminated from either the dorsal or the ventral side. Animals expressing channelrhodopsin were stimulated with a 60 s train (1 ms at 100 Hz) of blue light (415–475 nm) delivered using the same apparatus. We also used the same protocols on wild type and ChAT-EGFP cords to establish whether the light itself exerted any effects on the locomotor-like rhythm.

## Electrophysiological recordings and analysis

We recorded the activity of at least three lumbar ventral roots (L1, L2, L5, or L6) using tight-fitting plastic suction electrodes connected to a high gain amplifier. The signals were amplified (1000 X), band-pass filtered between 0.1 Hz and 3 KHz and digitized at 5 or 10 KHz (Digidata 1322A, 1440A; Molecular Devices Sunnyvale, CA) and stored for further analysis. When the green light was turned on or off, transient artifacts were introduced into the ventral root signals which were corrected manually using the adjust baseline feature of Clampfit (Molecular Devices). In addition, each blue light pulse introduced a small artifact in the ventral root recordings. These were reduced or eliminated by

averaging the artifacts for the duration of the trial for each root and then subtracting the averaged artifact from each artifact. Each trial lasted 3 min (60 s pre-light, 60 s in the light, 60 s post-light). 5 s at the beginning and at the ends of the recordings were removed to align all signals in all trials to the beginning of the light (length of the signal was set to 170 s). The recorded signals were then quantified using wavelet analysis originally introduced by *Mor and Lev-Tov (2007)* for electrophysiological analysis and implemented here with custom Matlab (Mathworks, Natick, MA) scripts (https://github.com/avinashpujala/Spectral-Analysis (*Pujala, 2017a*) and https://github.com/avinashpujala/General (*Pujala, 2017b*), *Pujala et al., 2016*; copies are archived at https://github.com/elifesciences-publications/Spectral-Analysis and https://github.com/elifesciences-publications/General respectively). We either analyzed the integrated neurograms or the slow locomotor drive potentials. The ventral root discharge was integrated by first low pass filtering the records at 200 Hz, followed by high pass filtering at 10 Hz, rectification and then low pass filtering at 5 Hz. The slow potentials were extracted by low pass filtering the ventral root signals at 200 Hz, followed by band pass filtering from 0.01 to 5 Hz. Wavelet spectrograms (3200 points along the time dimension) were then generated from 3 to 5 trials of optical stimulation (see below). From these spectra, we extracted time series for the frequency and phase between the bilateral flexor-dominated roots (L1/2) and between the ipsilateral flexor-dominated (L1/2) and extensor-dominated (L5/6) roots. As the wavelet analysis produces artifacts (edge effects), we clipped 10 s at the beginning and the end of the time series when plotting the results. For each experiment the time series were averaged. Whenever the frequency dropped to zero (before the light or during the light), the trial was discarded. This can occur if the power of the signal is too low resulting in a discontinuity in the time series. We then considered experiments that had a minimum of three valid trials to perform the analysis. From these, we calculated the mean value of the phase and frequency for every trial under control conditions (before the light was turned on). These variables were then normalized to the control level to compare their values across trials and between experiments. As the phase is already normalized, we subtracted the control average from every value and kept the absolute change in phase. For the frequency, we divided all points in time by the control average, subtracted one from it, to set the control value to an average of zero. The integrated neurograms were resampled at 19 Hz and resized to have the same time course and duration as the time series derived from the wavelet analysis. We also normalized the integrated neurograms to zero for the control period. The normalized data are plotted as percentage changes. We then averaged the integrated neurograms for all experiments. If the values in the averaged integrated neurograms go below zero it reflects a decrease in the motoneuron firing whereas when they are positive, it shows an increase in motoneuron firing.

To compare statistically the extracted time series between two conditions, we used a bootstrap procedure which allows the use of a t-test without any assumptions about the distribution of the variables (*Falgairolle and O'Donovan, 2015*). This allowed us to establish if each point in the experimental and control time series was statistically different . All the trial-averaged time series were used to generate the averaged time series for each variable (phase, frequency, integrated neurogram). These data comprised a matrix (3200 x number of experiments) for each variable. The bootstrap procedure was iterated 10,000 times. We then plotted one condition and color-coded the results of the t-test with green for $p<0.05$, yellow for $p<0.01$, orange for $p<0.001$ and red for $p<0.0001$.

We also determined the average value (over 10 s) of each variable before the light, at the beginning of the light, just before the light was turned off and just after the light was turned off. For the Channelrhodopsin experiments, the maximum change in frequency before during and after the light was calculated. To compare these values under the different conditions (light on/off – light status; animal type – genetic identity; genetic identity/drug treatment) we used a 2-way ANOVA with a two stage linear step up procedure of Benjamini, Krieger and Yekutieli post hoc test (Prism 7.0, GraphPad software, La Jolla, CA). The locomotor frequencies were calculated from the control part of the times series (before the light was on), and we compared them using a 1-way ANOVA with a two stage linear step up procedure of Benjamini, Krieger and Yekutieli post-test or a t-test for ChAT-ChR2. All results are given as mean ± SD. In all figures, *$p<0.05$, **$p<0.01$, ***$p<0.001$, ****$p<0.0001$.

## Intracellular recording from motoneurons

Microelectrodes were pulled from borosilicate capillaries with a microelectrode puller (model P-97; Sutter Instruments). Pipettes (2.5–7 MΩ) were filled with intracellular solution (NaCl 10 mM;

K-Gluconate 130 mM; MgCl$_2$ 1 mM; HEPES 10 mM; Na$_2$ATP 1 mM; EGTA 11 mM; CaCl$_2$ 0.1 mM; pH adjusted to 7.2–7.4 with KOH). The liquid junction potential was not corrected. After removing the dura and pia maters, blind recordings were obtained by targeting the motoneuron pools either ventrally or laterally. Two sets of experiments were performed using whole cell recordings. To assess the effect of hyperpolarizing motoneurons during the light, we recorded ten motoneurons which were identified by their antidromic spike in response to ventral root stimulation. We counted four additional cells as motoneurons even though they did not generate an antidromic spike. All four neurons were recorded in the ventral part of the cord near the motoneuron pool, and were hyperpolarized by the light. Three of the motoneurons received a small antidromically initiated short latency depolarization arising from antidromically-activated motoneurons that were electrically coupled to the recorded motoneuron. In the second set of experiments, we tested the effects of carbenoxolone (100 μM) on gap-junction coupling between motoneurons. We recorded from five antidromically-identified motoneurons in the L5 segment, in ChAT-Arch spinal cords. The ventral root was stimulated using duration of either 250 μs or 50 μs and the threshold for evoking the action potential determined. We then recorded the sub-threshold, short-latency depolarizations (averaging 20 trials for each short-latency depolarization) that represent the electrically-coupled action potentials from other antidromically activated motoneurons. The potentials were recorded at several membrane potentials to demonstrate their voltage-independence. The amplitude of short latency depolarizations was quantified following collision of the antidromic action potential with an orthodromically-evoked action potential (*Chang et al., 1999*). Measurements were made at 2x the threshold for the antidromic action potential. The amplitude of the coupling potential was obtained by subtracting the orthodromic action potential from the collided antidromic action potential. We then performed the same collision protocol in presence of carbenoxolone while monitoring the resistance of the cell. All recordings were obtained using a Multiclamp 700A amplifier (DC-3kHz, Molecular Devices). The input resistance of the cell was calculated from the slope of the current/ voltage (200 ms duration) plot within the linear range.

## Immunohistochemistry

The L2-L1 or L5-L6 segments of spinal cords were dissected and fixed in 4% paraformaldehyde for at least 4 hr at room temperature. They were then transferred to phosphate buffered saline (PBS), and embedded in a 5% Agar/PBS solution. Transverse sections (60 μm) were cut on V1000 vibratome (Leica Biosystems, Buffalo Grove, IL). The sections were blocked in 10% normal donkey serum in 0.01M PBS with 0.1% Triton X-100 for 90 min, and subsequently incubated overnight at room temperature in Chicken Anti-GPF antibody (ab13970, dilution 1:1000, Abcam, Cambridge, MA), Goat Anti-ChAT antibody (AB144P, dilution 1:100, Millipore, Temecula, CA). The following day, the sections were washed for an hour and then incubated for 3 hr in secondary antibodies: Donkey Anti-Chicken-FITC and Donkey Anti-Goat-Cy5 (dilution 1:50, Jackson ImmunoResarch, West Grove, PA). The sections were then mounted on slides and cover-slipped with a Glycerol/PBS solution (3:7). Images were acquired using a LSM510 Carl Zeiss confocal microscope with either 10X (air) and 40X (oil) objectives.

## Acknowledgements

The authors thank the intramural program of NINDS for financial support.

## Additional information

### Funding

| Funder | Grant reference number | Author |
| --- | --- | --- |
| National Institutes of Health | NINDS Intramural program | Melanie Falgairolle<br>Joshua G Puhl<br>Avinash Pujala<br>Wenfang Liu<br>Michael J O'Donovan |
| National Institutes of Health | NINDS Intramural NRSA | Joshua G Puhl |

The funders had no role in study design, data collection and interpretation, or the decision to submit the work for publication.

### Author contributions

MF, Conceptualization, Data curation, Software, Formal analysis, Validation, Investigation, Visualization, Methodology, Writing—original draft, Writing—review and editing; JGP, Funding acquisition, Investigation, Writing—review and editing; AP, Software, Investigation, Writing—review and editing; WL, Resources, Investigation, Visualization; MJO'D, Conceptualization, Resources, Data curation, Formal analysis, Supervision, Validation, Visualization, Methodology, Writing—original draft, Project administration, Writing—review and editing

### Author ORCIDs

Melanie Falgairolle, http://orcid.org/0000-0001-5243-4714
Michael J O'Donovan, http://orcid.org/0000-0003-2487-7547

### Ethics

Animal experimentation: All experiments were carried out in compliance with the National Institutes of Neurological Disorders and Stroke Animal Care and Use Committee (Animal Protocol Number 1267-12 and 1267-15).

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
