## [Decision Letter]

Thank you for submitting your article "Motoneurons regulate the central pattern generator during drug-induced locomotor-like activity in the neonatal mouse" for consideration by *eLife*. Your article has been favorably evaluated by Eve Marder (Senior Editor) and three reviewers, one of whom, Ole Kiehn, is a member of our Board of Reviewing Editors.

The reviewers have discussed the reviews with one another and the Reviewing Editor has drafted this decision to help you prepare a revised submission. We hope you will be able to submit the revised version within two months.

Summary:

In this study Falgairolle et al. investigate the possibility that motor neurons in the mouse spinal cord retrogradely influence the premotor rhythm- and pattern-generating spinal circuits. A synaptic retrograde signal mediated through acetylcholine release from motor neuron has previously been postulated for tadpole while experiments in zebrafish have shown that motor neurons are gap-junction coupled with premotor neurons that are possibly involved in rhythm-generation. Whether similar feedback mechanisms are present in the mammalian spinal cord has not been investigated in detail before. The present study used optogenetic manipulations in combination with pharmacology to block synaptic transmission and gap-junctions to directly test the contribution of motor neuron activity to rhythm- and pattern-generation. The study provides compelling evidence that suppressing or activating motor neuron activity during ongoing locomotor-like activity can affect both the pattern and rhythm, suggesting that motor neurons provide a retrograde signal not only to Renshaw cells and other motor neurons (which is well known) but also to premotor locomotor circuits. Unlike in aquatic animals, the retrograde signal is not mediated through acetylcholine release or gap-junctions but through glutamatergic neurotransmission from motor neurons to pro-motor circuits.

The study provides new insight to the function of spinal locomotor networks and provides a mechanistic explanation for the release of glutamate from central motor neuron collaterals that was discovered almost 15 years ago by this group and others. However, there are limitations in the study that prevent this conclusion from being solid although the evidence might be strong. To address these problems further experimental evidence is needed as outlined below in addition to other concerns that need to be addressed to strengthen the conclusion.

Essential revisions:

1) An important caveat to the work is the lack of motor neuron specific markers that can be used to drive expression of opsins in motor neurons alone. The authors acknowledge this limitation and utilize two different markers, ChAT which is expressed by motor neurons as well as cholinergic interneurons and preganglionic neurons, and Isl1, which is expressed by motor neurons as well as primary afferents, dI3 interneurons, and sympathetic motor neurons. They find that the effects of opsin-mediated inhibition appear similar when the expression was driven by ChAT or Isl1 and use both the similarities in effect and pharmacology to exclude the possibility that the effect of optogenetic stimulation is due to stimulation of other cells than the motor neurons. Although these data are used to support a primary role of motor neurons in the observed effects, this is based on the assumption that motor neurons, but not other spinal cholinergic neurons, co-release substances other than acetylcholine.

All the reviewers express concerns about the lack of selectivity of the Cre-lines but also acknowledge that presently there are no Cre-lines that are completely motor neuron selective. In light of that, we feel that we cannot request new experiments to use other lines than those already used. However, it remains important to consider and discuss this caveats in more detail.

A) Consider the literature that has been published on some of the non-MN classes e.g. the Pitx2 and the DI3 cells. Discuss if stimulation or elimination of these categories of cells have effect on locomotion per se.

B) Similarly, what is known about preganglionic cells and their effect of locomotion?

C) Liu et al. 2009 show co-localization of glutamate and Ach in ventral interneurons. This should be acknowledged because synaptic transmission from such cells will be a confounding factor for the optogenetic experiments.

D) Show dorsal side up light-stimulation in Chat animal that presumably leave motor neurons untouched. To this end varying the light-intensity would also be informative.

E) It might be worth evaluating a dorsal horizontal sectioning ad modum Dougherty and Kiehn 2010 in the Isl1 prep which should potentially remove the Ist1 positive interneurons leaving only the motor neurons.

2) The data analysis is not well described. For example, the authors often refer to changes in "motor neuron firing" referring to "Int. Change" in figures. However, the exact parameter being measured is not clear. One assumes the authors are measuring the amplitude or area under the curve of rectified/integrated traces of ventral root output? If so, it is sometimes hard to see the measured change in moto neuron output (Int. Change) in raw traces. For example, in Figure 2 a decrease in "Int Change (%)" is reported but it is hard to see any decrease in the amplitude of bursts in Figure 2? This is an issue for reported measures of "motor neuron firing" throughout the manuscript. It would have been nice to see clearer effects of light-mediated inhibition (or activation) on the firing output of individual motor neurons as verification of the optogenetic methodology employed. We suggest that the authors consistently use the wording: amplitude modulation, frequency, and phase relationship.

3) Why are you stimulating for 60s? It seems unnecessary to stimulate for so long to see an effect. It is puzzling that you see immediate effects that vanish over time and then followed by a rebound. Please explain.

4) The effect seen in Figure 4 and in some but not all the figures of root recordings is puzzling. There is a complete lack of rhythmicity right after the onset of inhibition which the authors do not contribute to a 'flattening' caused by the motor neurons being close to the reversal potential for chloride (and a dominating inhibition). This is quite strange because a complete lack of rhythm would mean that the motor neurons are the only rhythm-generating neurons, which we don't think that the authors propose. Some explanation for this phenomenon is essential. To exclude that the effect is due to inhibition of intrinsic motor neuron properties by hyperpolarization, that will look like a pre-synaptically mediated frequency change the author should perform experiments where they shine light to a restricted part of the spinal cord (L4-L5 or L2-L3) while recording from MNs in a segment that is not exposed to light (and also record from several VR). It will important to show that the change in rhythm and/or pattern are seen in ventral root and/or motor neurons that are not exposed to light. These experiments are essential and will clearly exclude that some of the effects reported are due to direct manipulation of intrinsic motor neurons properties.

5) The light-induced changes in locomotor pattern in WT mice are in all likelihood due to heat generated by the diode. Did the authors try to use other light sources? Since the heat-induced changes went in the opposite direction than the 'biological' we think it is okay. But some experiments without heat-induced changes would have been advantageous. Check temperature in bath.

6) The 'noise level' of the DC recordings consistently change during the opsin inhibition. What is the mechanism for this phenomenon?

7) Carbenoxelone is a dirty drug with variable effects on gap junctions. Did the authors provide evidence that it actually blocked gap junction coupling in this prep? Some sort of evidence – motor neuron recordings and antidromic activation would be very helpful (e.g. Tresh and Kiehn 2000).

8) The Discussion is very long and unstructured. It needs to be focused on the key finding with structured discussion of the caveats in Methods and interpretations. A discussion of the functional need for retrograde signal could be expanded – when will motor neurons be strongly activated to change frequency of locomotion – as well as considerations of why the mechanisms of a retrograde signal might be different across phyla and the efficiency of the different components.

[Editors' note: further revisions were requested prior to acceptance, as described below.]

Thank you for submitting your article "Motoneurons regulate the central pattern generator during drug-induced locomotor-like activity in the neonatal mouse" for consideration by *eLife*. Your article has been reviewed by two peer reviewers, and the evaluation has been overseen by a Reviewing Editor and Eve Marder as the Senior Editor. The reviewers have opted to remain anonymous.

The reviewers have discussed the reviews with one another and the Reviewing Editor has drafted this decision to help you prepare a revised submission.

This manuscript has been extensively revised with added clarifying experiments and text editing. The authors should be complemented for this great work.

In particular:

1) The authors are now providing a discussion of the lack of motor neuron specific markers and the possible involvement of Chat-expressing interneurons in the phenotype they observe.

2) They have added dorsal shaved preparations to the study that leaves removes the cell bodies of dI3 from the cord. They obtain the same phenotype, supporting a contribution from motor neurons.

3) They have provided a dose response curve.

4) They have done experiments with local stimulation that leaves some motor neurons unaffected which shows that the effect on frequency is not only due to a selective interference with motor neuron properties.

5) They have clarified the effect in the native spinal cord and ascribed it to heat.

6) The have now provided direct evidence that carbenoxelone blocks gap junction coupling among motor neurons although just partially.

7) They have improved the Discussion taking into account the caveats and added a comparison with other species.

Together the revision adds a substantial amount of work that has strengthened the case.

There is one major issue that remains to be addressed in writing in a final revision.

1) The complete lack of rhythmicity right after the onset of inhibition which is seen in Figure 4 and also in some – but not all – of the new motor neuron recordings remains puzzling. The authors go through a long discussion that it is not due to a change in the locomotor drive potential either in flexor or in extensor motor neurons. These arguments make sense but then it means that the rhythm is lost – or at least that the synaptic drive to motor neurons is lost. How is this possible? This question was posed to the authors, but the answer is unclear. It means that losing the synaptic excitatory drive from motor neurons has such a strong effect on the rhythm generation that it stops (or that the pattern generating inhibitory and excitatory layer is blocked). Is this what the authors suggest? – If so the authors should state this much more clearly in the text because this is a dramatic effect that goes much beyond just providing positive feedback from the motor neurons to the rhythm-generating layer and changing its frequency. If not they need to provide an alternative explanation for this effect that stands out so clearly in the intracellular recordings.

A clarification of this issue should be incorporated in the section "functional need for retrograde signal" which could be strengthened by a clearer statement about the functional role/physiological significance of such positive feedback?

---

## [Author Response]

*Essential revisions:*

*1) An important caveat to the work is the lack of motor neuron specific markers that can be used to drive expression of opsins in motor neurons alone. The authors acknowledge this limitation and utilize two different markers, ChAT which is expressed by motor neurons as well as cholinergic interneurons and preganglionic neurons, and Isl1, which is expressed by motor neurons as well as primary afferents, dI3 interneurons, and sympathetic motor neurons. They find that the effects of opsin-mediated inhibition appear similar when the expression was driven by ChAT or Isl1 and use both the similarities in effect and pharmacology to exclude the possibility that the effect of optogenetic stimulation is due to stimulation of other cells than the motor neurons. Although these data are used to support a primary role of motor neurons in the observed effects, this is based on the assumption that motor neurons, but not other spinal cholinergic neurons, co-release substances other than acetylcholine.*

*All the reviewers express concerns about the lack of selectivity of the Cre-lines but also acknowledge that presently there are no Cre-lines that are completely motor neuron selective. In light of that, we feel that we cannot request new experiments to use other lines than those already used. However, it remains important to consider and discuss this caveats in more detail.*

*A) Consider the literature that has been published on some of the non-MN classes e.g. the Pitx2 and the DI3 cells. Discuss if stimulation or elimination of these categories of cells have effect on locomotion per se.*

We have now added a discussion of both cell classes, their roles in locomotor-like activity, and the effects of silencing them on the locomotor-like rhythm.

*B) Similarly, what is known about preganglionic cells and their effect of locomotion?*

We now discuss the possible contribution of preganglionic neurons to locomotor rhythmogenesis.

*C) Liu et al. 2009 show co-localization of glutamate and Ach in ventral interneurons. This should be acknowledged because synaptic transmission from such cells will be a confounding factor for the optogenetic experiments.*

The confounding possibility that some cholinergic interneurons release glutamate in addition to acetylcholine has been considered in the Discussion. We explicitly cite the work of Liu et al., (2009) and discuss the extent to which the existence of such neurons could compromise our conclusions.

*D) Show dorsal side up light-stimulation in Chat animal that presumably leave motor neurons untouched. To this end varying the light-intensity would also be informative.*

We have now performed these experiments. We used light intensities varying from 20 to 100% of the maximum light intensity, and compared the effects of illuminating the cord from the dorsal or the ventral surface. We found that as the light intensity decreased the suppression of motoneuron firing was reduced and the changes in the frequency of the locomotor rhythm were smaller and more variable (see new Figure2—figure supplement 2). When the light intensity was reduced to 20-25% of the maximum motoneurons firing was reduced by ~25-30% and this was similar whether the cord was illuminated ventrally or dorsally. Therefore, we were not able to use dorsal illumination without depressing motoneuron activity, probably because of light reaching motoneuron dendrites.

*E) It might be worth evaluating a dorsal horizontal sectioning ad modum Dougherty and Kiehn 2010 in the Isl1 prep which should potentially remove the Ist1 positive interneurons leaving only the motor neurons.*

We have performed this experiment in 2 islet1 cords as suggested and find that the results are similar to the effects of light on the intact cord. These results are presented in a new figure (Figure 7—figure supplement 1).

*2) The data analysis is not well described. For example, the authors often refer to changes in "motor neuron firing" referring to "Int. Change" in figures. However, the exact parameter being measured is not clear. One assumes the authors are measuring the amplitude or area under the curve of rectified/integrated traces of ventral root output? If so, it is sometimes hard to see the measured change in moto neuron output (Int. Change) in raw traces. For example, in Figure 2 a decrease in "Int Change (%)" is reported but it is hard to see any decrease in the amplitude of bursts in Figure 2? This is an issue for reported measures of "motor neuron firing" throughout the manuscript. It would have been nice to see clearer effects of light-mediated inhibition (or activation) on the firing output of individual motor neurons as verification of the optogenetic methodology employed. We suggest that the authors consistently use the wording: amplitude modulation, frequency, and phase relationship.*

We used the integrated neurograms as a measure of motoneuron firing. During drug-induced locomotor-like activity, the ventral roots show continuous firing, because none of them are purely flexor or extensor, but with the largest burst associated with the dominant motoneuron type in that root. That implies that the integrated neurograms have a baseline that is greater than zero in control condition. For the ChAt-Arch spinal cords, this baseline activity and the amplitude of the burst of motoneuron firing are reduced, however the amplitude of the slow potentials may not be reduced and indeed often increases in the flexor segments. To illustrate these effects more clearly, we have now included figures of high-pass filtered neurograms and superimposed integrated neurograms for ChAT-Arch (Figure 4—figure supplement 3) and ChAT-ChR2 (Figure 5—figure supplement 2) experiments. We also have quantified the effect of light on the firing of individual motoneurons in the ChAT-Arch cords. These data are illustrated a new supplementary figure showing the effect of light on the firing of 4 flexor and 8 extensor motoneurons in the ChAT-Arch cords.

*3) Why are you stimulating for 60s? It seems unnecessary to stimulate for so long to see an effect. It is puzzling that you see immediate effects that vanish over time and then followed by a rebound. Please explain.*

We used a pre-light, light and post-light duration of 60 seconds to obtain enough cycles for the wavelet-based analysis. As shown in the intracellular recordings in the ChAT-Arch spinal cords, the strongest effect of the light happens during the first 20 seconds as the membrane potential of motoneurons slowly recover over time. In some examples the neurons resume spiking while still being hyperpolarized by the light. When the light is subsequently turned off, motoneurons get often depolarized triggering a rebound in spiking. We have added a discussion of these issues in the text.

*4) The effect seen in Figure 4 and in some but not all the figures of root recordings is puzzling. There is a complete lack of rhythmicity right after the onset of inhibition which the authors do not contribute to a 'flattening' caused by the motor neurons being close to the reversal potential for chloride (and a dominating inhibition). This is quite strange because a complete lack of rhythm would mean that the motor neurons are the only rhythm-generating neurons, which we don't think that the authors propose. Some explanation for this phenomenon is essential.*

We interpret the transient lack of rhythmicity seen in some neurons and some ventral root recordings immediately following light onset as a slowing of the rhythm. This point has been clarified in the text.

*To exclude that the effect is due to inhibition of intrinsic motor neuron properties by hyperpolarization, that will look like a pre-synaptically mediated frequency change the author should perform experiments where they shine light to a restricted part of the spinal cord (L4-L5 or L2-L3) while recording from MNs in a segment that is not exposed to light (and also record from several VR). It will important to show that the change in rhythm and/or pattern are seen in ventral root and/or motor neurons that are not exposed to light. These experiments are essential and will clearly exclude that some of the effects reported are due to direct manipulation of intrinsic motor neurons properties.*

We have now performed these experiments exposing the light to the rostral segments only (T13-L4) and find that the change in frequency is still present in the L5 segment even though the firing of those motoneurons is minimally affected (new Figure 4—figure supplement 3). This indicates that modification of the intrinsic properties of the L5 motoneurons is not responsible for their change in frequency.

*5) The light-induced changes in locomotor pattern in WT mice are in all likelihood due to heat generated by the diode. Did the authors try to use other light sources? Since the heat-induced changes went in the opposite direction than the 'biological' we think it is okay. But some experiments without heat-induced changes would have been advantageous. Check temperature in bath.*

We chose the light source described in the paper because it illuminated the whole lumbar cord. We tried other light sources before settling on this one, but they did not suppress motoneuron discharge satisfactorily. We measured the temperature locally (right under the light) in the bath, the green light increased the temperature by about 2 to 3 °C while the blue light increased it by about 0.5°C.

*6) The 'noise level' of the DC recordings consistently change during the opsin inhibition. What is the mechanism for this phenomenon?*

We think that the reviewer is referring to the spiking between the bursts. As we described above during drug-induced locomotor-like activity, the ventral roots show continuous firing, because none of them are purely flexor or extensor. Suppression of firing in the non-dominant population may appear as a reduction in ‘noise’ level. This change during the light is because the efficacy of the light at suppressing discharge declines during the 60 seconds of light exposure.

*7) Carbenoxelone is a dirty drug with variable effects on gap junctions. Did the authors provide evidence that it actually blocked gap junction coupling in this prep? Some sort of evidence – motor neuron recordings and antidromic activation would be very helpful (e.g. Tresh and Kiehn 2000).*

We have now performed these experiments and included them in Figure 8.

*8) The Discussion is very long and unstructured. It needs to be focused on the key finding with structured discussion of the caveats in Methods and interpretations. A discussion of the functional need for retrograde signal could be expanded – when will motor neurons be strongly activated to change frequency of locomotion – as well as considerations of why the mechanisms of a retrograde signal might be different across phyla and the efficiency of the different components.*

The discussion has been changed according to the reviewers’ suggestions.

[Editors' note: further revisions were requested prior to acceptance, as described below.]

*[…] There is one major issue that remains to be addressed in writing in a final revision.*

*1) The complete lack of rhythmicity right after the onset of inhibition which is seen in Figure 4 and also in some – but not all – of the new motor neuron recordings remains puzzling. The authors go through a long discussion that it is not due to a change in the locomotor drive potential either in flexor or in extensor motor neurons. These arguments make sense but then it means that the rhythm is lost – or at least that the synaptic drive to motor neurons is lost. How is this possible? This question was posed to the authors, but the answer is unclear. It means that losing the synaptic excitatory drive from motor neurons has such a strong effect on the rhythm generation that it stops (or that the pattern generating inhibitory and excitatory layer is blocked). Is this what the authors suggest? – If so the authors should state this much more clearly in the text because this is a dramatic effect that goes much beyond just providing positive feedback from the motor neurons to the rhythm-generating layer and changing its frequency. If not they need to provide an alternative explanation for this effect that stands out so clearly in the intracellular recordings.*

*A clarification of this issue should be incorporated in the section "functional need for retrograde signal" which could be strengthened by a clearer statement about the functional role/physiological significance of such positive feedback?*

We have added the following to the section entitled “Function of motoneuron feedback to the CPG” in the Discussion.

“One possibility is that the feedback from motoneurons constitutes an efference copy of the motoneuron output that could be compared to ongoing motor commands and relayed to higher centers including the cerebellum (see (Wolpert and Miall, 1996) for a discussion of internal models of the motor system). […] We are not suggesting that the excitatory input from motoneurons is essential for the operation of the CPG but that it may be important in regulating the excitability of the CPG.”

And to the section entitled “Mechanism of the light-induced changes in the locomotor-like rhythm and the post-light rebound” in the Discussion.

“One possibility is that the recovery of the locomotor frequency that occurs during the light reflects a homeostatic compensation that is mediated by an increase in the excitatory interneuronal input to the CPG. When the light is turned off, the motoneuronal input is restored and together with the increased interneuronal input results in the transient increase in the locomotor frequency.”